Ma *et al. Genome Biology*    (2022) 23:208

**METHOD**

# Population structure discovery in meta-analyzed microbial communities and inflammatory bowel disease using MMUPHin

Siyuan Ma[1], Dmitry Shungin[2], Himel Mallick[2], Melanie Schirmer[2], Long H. Nguyen[3], Raivo Kolde[2], Eric Franzosa[1], Hera Vlamakis[2], Ramnik Xavier[2*] and Curtis Huttenhower[1*]

*Correspondence:
xavier@molbio.mgh.harvard.edu;
chuttenh@hsph.harvard.edu

[1] Harvard Chan Microbiome
in Public Health Center, Harvard
T.H. Chan School of Public
Health, Boston, MA, USA
[2] Broad Institute of MIT
and Harvard, Cambridge, MA,
USA
Full list of author information is
available at the end of the article

## Abstract

Microbiome studies of inflammatory bowel diseases (IBD) have achieved a scale for meta-analysis of dysbioses among populations. To enable microbial community meta-analyses generally, we develop MMUPHin for normalization, statistical meta-analysis, and population structure discovery using microbial taxonomic and functional profiles. Applying it to ten IBD cohorts, we identify consistent associations, including novel taxa such as *Acinetobacter* and *Turicibacter*, and additional exposure and interaction effects. A single gradient of dysbiosis severity is favored over discrete types to summarize IBD microbiome population structure. These results provide a benchmark for characterization of IBD and a framework for meta-analysis of any microbial communities.

**Keywords:** Inflammatory bowel disease, Metagenomics, Dysbiosis, Meta-analysis, Batch effect

## Background

Meta-analysis for molecular epidemiology in large populations has seen great success in linking high-dimensional 'omic features to complex health-related phenotypes. One example of this is in genome-wide association studies (GWAS [1]), where the appropriate study scale, achieved by rigorous integration of multiple cohorts, has both facilitated reproducible discoveries (in the form of disease-associated loci [2–4]) and addressed confounding due to unobserved population structure [5]. The inflammatory bowel diseases (IBD) represent a particular success story for GWAS meta-analysis [3, 4], and environmental and microbial contributors complementing the condition's complex genetic architecture have been detailed by many individual studies [6–8]. However, in the absence of methods appropriate for large-scale microbial meta-analysis, the extent to which these findings reproduce across studies, or can be extended by increased joint

sample sizes, remains undetermined. Likewise, it is unclear whether reproducible population structure in the microbiome, such as microbially driven IBD "subtypes," exists to help explain the clinical heterogeneity of these conditions [9].

Meta-analysis of microbial community profiles presents unique quantitative challenges relative to other types of 'omics data such as GWAS [10] or gene expression [11]. These include particularly strong batch, inter-individual, and inter-population differences, and statistical issues including zero inflation and compositionality [12, 13]. Consequently, methods to correct for cohort and batch effects from other 'omics settings [14–17] are not directly appropriate. Two recent studies have suggested quantile normalization [18] and Bayesian Dirichlet-multinomial regression (BDMMA) [19] for microbial profiles, which are applicable to a limited subset of differential abundance tests and do not provide batch-corrected profiles. To date, there are no methods permitting the joint analysis of batch-corrected microbial profiles for most study designs.

IBD represents one of the best-studied, microbiome-linked inflammatory phenotypes to date which thus stands to benefit from such approaches [20, 21]. Among the inflammatory bowel diseases, Crohn's disease (CD) and ulcerative colitis (UC) have been individually linked with structural and functional changes in the gut microbiome in many individual studies [21]. Each of CD and UC can itself be highly heterogeneous within the IBD population, however, and diversity in disease-associated gut microbial features has not been consistently associated with factors including disease subtype, progression, or treatment response [7, 9, 22, 23]. Of note, two meta-analysis studies included IBD as one of several phenotypes [24, 25]. These studies were not IBD-specific, did not have access to appropriate normalization techniques, nor took the aforementioned factors into account. The complexity of microbial involvement in IBD, and the presence of substantial unexplained variation in the manifestation of its symptoms, makes it particularly appropriate for application of meta-analysis techniques.

In this work, we introduce and validate a statistical framework for population-scale meta-analysis of microbiome data, and apply it to the largest collection to date of ten published 16S rRNA gene sequencing-based IBD studies (Table 1) to identify consistent disease associations and population structure. We found both previously documented and novel microbial links to the disease, with further differentiation among subtypes, phenotypic severity, and treatment effects. We further confidently conclude that there are no apparent, reproducible microbiome-based subtypes within CD or UC, which are instead a population structure gradient from less to more "pro-inflammatory" ecological configurations. Our work thus represents one of the first large-scale efforts to assess consistency in gut microbial findings for IBD and provides methodology supporting future microbial community meta-analyses.

## Results

### Integrating 10 studies of the IBD stool and mucosal microbiomes

We collected and uniformly processed ten published 16S studies of the IBD gut microbiome (Table 1, Fig. 1a, Additional file 1: Figs. S1-S4, Additional file 2: Supplemental Notes, Additional file 3: Table S1) totaling 2179 subjects and 5151 samples. These studies range widely in terms of cohort designs and population characteristics, including recent-onset and established disease patients, cross-sectional and

**Table 1** Ten uniformly processed 16S rRNA gene sequencing studies of the IBD mucosal/stool microbiomes. For longitudinal cohorts, numbers in parentheses indicate baseline sample size. For age, mean and standard error (parenthesized) are shown. Additional covariates are summarized in Additional file 3: Table S1

| Study | Brief description | N subject | N sample | Phenotype(s) | Age | Gender | Sample type(s) |
|---|---|---|---|---|---|---|---|
| PROTECT [23] | Longitudinal cohort of newly diagnosed UC | 405 | 1212 (539) | UC 405 | 12.71 (3.29) | Male 52%/Female 48% | Biopsy 22%/Stool 78% |
| RISK [7] | Pediatric cohort of treatment-naïve CD | 631 | 882 | CD 430/Control 201 | 12.16 (3.22) | Male 59%/Female 41% | Biopsy 72%/Stool 28% |
| Herfarth [26] | Densely (daily) sampled longitudinal cohort | 31 | 860 (31) | CD 19/Control 12 | 36.03 (14.12) | Male 35%/Female 58%/Missing 6% | Stool |
| Jansson-Lamendella [22] | Longitudinal follow-up with fecal samples | 137 | 683 (137) | CD 49/UC 60/Control 28 | | Male 42%/Female 58% | Stool |
| Pouchitis [27] | Patients recruited underwent IPAA for treatment of UC or FAP prior to enrollment. | 353 | 577 | CD 42/UC 266/Control 45 | 46.19 (13.58) | Male 52%/Female 48% | Biopsy |
| CS-PRISM [28] | Cross-sectional cohort nested in PRISM | 397 | 467 | CD 215/UC 144/Control 38 | 41.68 (15.22) | Male 47%/Female 53% | Biopsy 29%/Stool 71% |
| HMP2 [9] | Large cohort of newly diagnosed IBD with multi-'omics measurement. | 81 | 177 (162) | CD 37/UC 22/Control 22 | 29.76 (19.63) | Male 51%/Female 49% | Biopsy |
| MucosalIBD [29] | Pediatric cohort with Paneth cell phenotypes | 83 | 132 | CD 36/Control 47 | 12.93 (3.65) | Male 58%/Female 42% | Biopsy |
| LSS-PRISM [30] | Longitudinal cohort nested in PRISM. | 18 | 88 (19) | CD 12/UC 6 | 30.37 (10.52) | Male 39%/Female 61% | Stool |
| BIDMC-FMT [31] | FMT Trial design | 8 | 16 | CD 8 | 38.38 (12.73) | Male 62%/Female 38% | Stool |

longitudinal sampling, pediatric and adult populations, diseases (CD and UC), treated and treatment-naive patients, biopsy and stool samples, and inclusion of healthy/non-IBD controls. Covariates were manually curated to ensure consistency across studies ("Methods"). Major factors available from all or most studies included demographics (age/sex/race), biogeography, disease location and/or extent, antibiotic usage, immunosuppression, and steroid and/or 5-ASA usage.

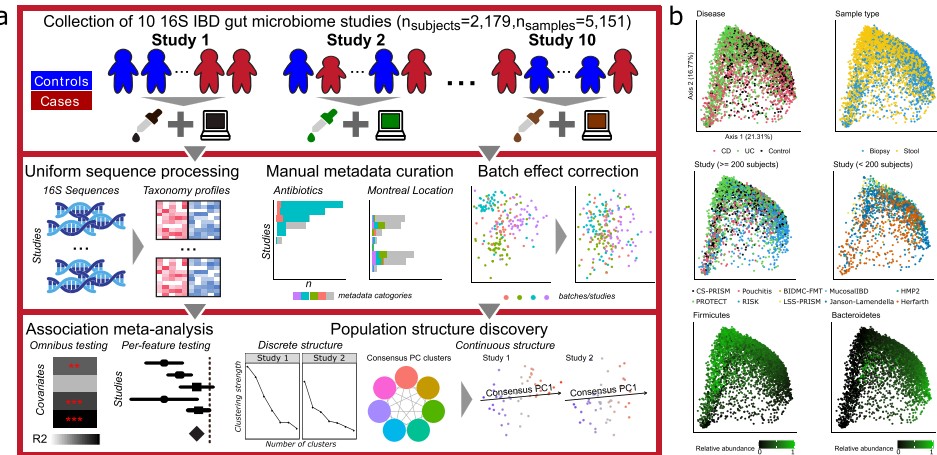

**Fig. 1** A method for large-scale microbial community meta-analysis and its application to inflammatory bowel disease. **a** We developed a novel statistical framework, MMUPHin, allowing joint normalization and meta-analysis of large microbial community profile collections with heterogeneous and complex designs (multiple covariates, longitudinal samples, etc.). We applied it to a collection of 10 inflammatory bowel disease studies comprising 2179 subjects and 5151 total samples (Table 1). We uniformly processed the associated sequence data and harmonized metadata across cohorts. Microbial taxonomic profiles were then corrected for batch and study effects before downstream analyses for omnibus and per-feature association with disease phenotypes and unsupervised population structure discovery. **b** MDS ordination of all microbial profiles (Bray-Curtis dissimilarity) before batch correction visualize the strongest associations with gut microbial composition, including disease, sample type (biopsy or stool), cohort (visualized separately for larger and smaller studies), and dominant phyla

Using this joint dataset and upon uniform bioinformatics processing ("Methods"), we first assessed the factors that corresponded to overall variation in microbiome structure, which included disease status, sample type (biopsy versus stool), and dominant phyla (Bacteroidetes and Firmicutes, Fig. 1b). Cohort effects prior to batch correction and meta-analysis were also significant. Microbiome differences associated with disease were notable even without normalization. However, this can be misleading due to the confounding of cohort structure between studies, such as the differentiation between RISK (a predominantly mucosal study of CD) and PROTECT (a predominantly stool study of UC). Inter-individual differences largely independent of population or disease, such as Bacteroidetes versus Firmicutes dominance, were also universal among studies and sample types as expected [9, 32]. Many of these factors were of comparable effect size, both visually and as quantified below, emphasizing the need for covariate-adjusted statistical modelling to delineate the biological (disease, treatment) and technical (cohort, batch) effects associated with individual taxa throughout the cohorts (Additional file 1: Figs. S1-S4, Additional file 2: Supplemental Notes).

### A statistical framework for meta-analysis of microbial community profiles

We developed a collection of novel methods for meta-analysis of environmental exposures, phenotypes, and population structures across microbial community studies, specifically accounting for technical batch effects and interstudy differences ("Methods," Fig. 1a). Jointly named MMUPHin (*M*eta-Analysis *M*ethods with a *U*niform *P*ipeline for *H*eterogeneity *in* microbiome studies), our methods consist of three main components: batch and study effect correction (MMUPHin_Correct), meta-analyzed

differential abundance testing (MMUPHin_MetaDA), and population structure discovery (MMUPHin_Discrete and MMUPHin_Continuous). First, MMUPHin_Correct performs batch correction of microbial abundance data, by extending methods from the gene expression literature (ComBat [15]) to zero-inflated microbiome sequencing profiles. Based on linear modelling, the method can differentiate between technical effects (batch, study) versus covariates of biological interest (exposure, phenotype). Second, MMUPHin_MetaDA performs meta-analytical testing of per-feature (taxon, gene, or pathway) differential abundance effects, by combining well-validated data transformation and linear modelling combinations for microbial community profiles [33] with fixed and random effect modelling [34]. Lastly, MMUPHin_Discrete and MMUPHin_Continuous perform unsupervised discovery and validation of both discrete and continuous population structures in microbial community data (Additional file 1: Fig. S5). This is generalized from our previous approaches in cancer transcriptional subtyping [35]. Our methods are available as an R package [36] through Bioconductor [37].

### Comprehensive validation of MMUPHin via realistic synthetic data

We validated MMUPHin both in comparison to existing methods and through extensive simulation studies (Fig. 2), with simulated realistic microbial abundance profiles at different data dimensionality, biological/technical batch signal strength, and discrete/continuous population structures ("Methods," Additional file 1: Figs. S6-S10, Additional file 4: Table S2). These simulations were designed to be complementary to our application to and assessment of the IBD microbiome as described below, since they allow analysis of a controlled ground truth of outcome-associated and null microbial elements that is lacking in uncontrolled population settings. As detailed in "Methods," our simulation approach (a) generates realistic microbial profiles, so that the evaluation findings are generalizable to the appropriate target populations, and (b) is neutral to the evaluated methods (ComBat, quantile normalization, MMUPHin, etc.).

MMUPHin_Correct successfully reduced variability attributable to technical effects in simulated microbial profiles, as first quantified by the PERMANOVA R2 statistic [38] (Fig. 2a, b, Additional file 1: Fig. S6). This was true both in terms of reducing the overall microbial variability attributable to technical artifacts and in terms of the ratio of "biological" versus technical variability (Fig. 2a). ComBat correction [15], suited for gene expression data, was capable of reducing batch effects to a lesser degree, but also tended to reduce desirable "biological" variation in the process, likely due to noise introduced by it changing many zero counts to non-zero values. Previously proposed techniques for microbial community data, namely quantile normalization [18] and BDMMA [19], are only appropriate for differential abundance analysis and do not provide batch-normalized profiles, thus precluding PERMANOVA batch effect quantification; their per-feature testing performance is evaluated together with MMUPHin_MetaDA instead. MMUPHin_Correct thus provides batch-corrected microbial community profiles that retain biologically meaningful variation more than (or not even possible using) existing methods. Subsequently, for differential abundance testing, MMUPHin_MetaDA successfully corrected for false associations when batch/cohort effects were confounded with variables of interest, which is a common concern for 'omics meta-analysis [39],

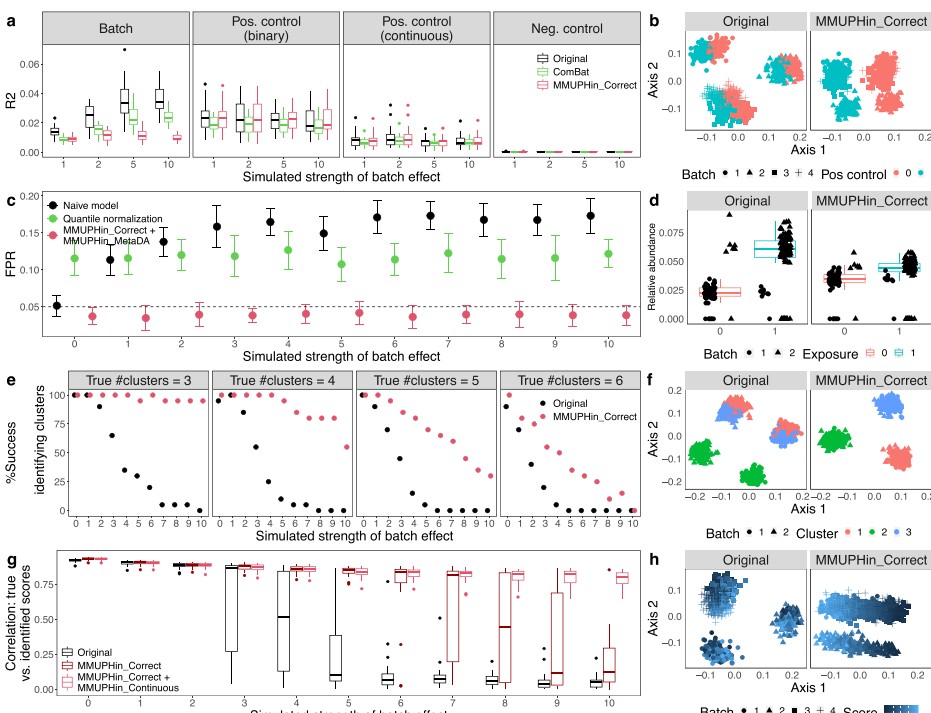

**Fig. 2** Effectiveness of batch correction, association meta-analysis, and unsupervised population structure discovery methods. All evaluations use simulated microbial community profiles as detailed in "Methods." Left panels summarize representative subsets of results (full set of simulation cases presented in Additional file 4: Table S2 and results in Additional file 1: Figs. S6-S9), and right panels show examples of batch-influenced data pre- and post-correction. **a**, **b** MMUPHin_Correct is effective for covariate-adjusted batch effect reduction while maintaining the effect of positive control variables. For panel **a**, PERMANOVA $R^2$ statistics summarize the effect of batch and positive/negative control variables on the overall microbial composition, before and after batch correction. Results shown correspond to the subset of details in Additional file 1: Fig. S6 with number of samples per batch = 500, number of batches = 4, and number of features = 1000 with 5% spiked with associations. **c**, **d** Batch correction and meta-analysis with MMUPHin_MetaDA reduces false positives when an exposure is spuriously associated with microbiome features due to an imbalanced distribution between batches. Corresponds to Additional file 1: Fig. S7 with number of samples per batch = 500, number of features = 1000 with 10% spiked associations, and case proportion difference between batches = 0.8. Evaluations of BDMMA generate low FPRs due to the zero-inflated nature of simulated microbial abundances, and are included only in Additional file 1: Fig. S7. **e**, **f** Batch correction improves correct identification of the true underlying number of clusters during discrete population structure discovery. Success rate is measured as the percentage of selecting the true number of clusters (before and after correction) across simulation iterations. Corresponds to Additional file 1: Fig. S8 with number of samples per batch = 500 and number of batches = 2. **g**, **h** Continuous structure discovery with MMUPHin_Continuous accurately recovers microbiome compositional gradients in a simulated population. We compare identified continuous structure loading with true scores with Pearson correlations. Corresponds to Additional file 1: Fig. S9 with number of batches = 6

while quantile normalization [18] and BDMMA [19] had either inflated or overly conservative false positive rates (Fig. 2c, d, Additional file 1: Figs. S7-S8).

We also validated MMUPHin's support for unsupervised population structure discovery, in addition to these "supervised" differential abundance and statistical association tests. In microbial communities, valid, generalizable population structure can manifest as either discretely clustered subtypes [40] or as continuously variable gradients of community configurations [41], but methods for discovery are particularly susceptible to false positives in the presence of technical artifacts [32, 41]. To this end, for

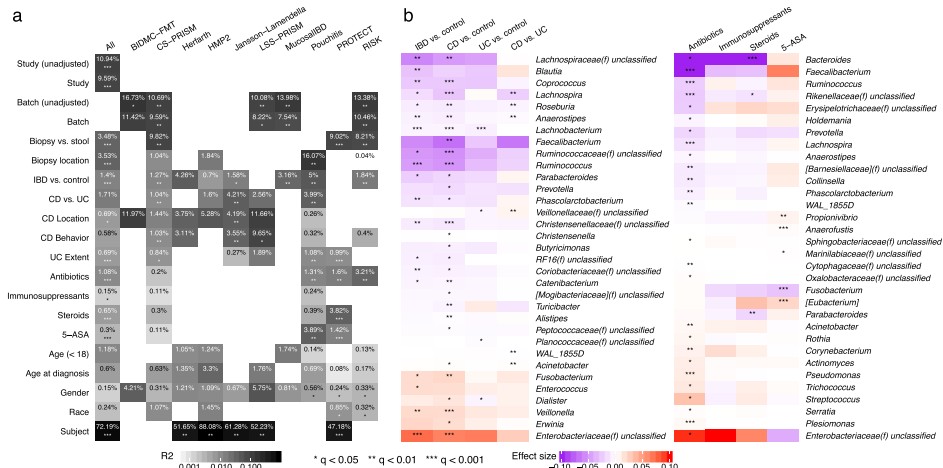

**Fig. 3** Meta-analytic omnibus and per-feature testing reveal novel and previously documented IBD associations. **a** Omnibus testing (PERMANOVA on Bray-Curtis dissimilarities with stratification and covariate control where appropriate, see "Methods" and Additional file 5: Table S3) identified between-subject differences as the greatest source of microbiome variability, with IBD phenotype, disease (CD/UC), and sample type (stool/biopsy) as additional main sources of biological variation. MMUPHin successfully reduced between-cohort and within-study batch effects, although these technical sources also remained significant contributors to variability. **b** Individual taxa significantly associated with IBD phenotypes or treatments after meta-analysis. Taxa are arranged by family-level median effect size of IBD vs. control for disease results and that of antibiotic usage for treatment results. Effect sizes are aggregated regression coefficients (across studies with random effects modelling) on arcsin square root-transformed relative abundances. Detailed model information in "Methods" and Additional file 5: Table S3. Individual study results in Additional file 6: Table S4

discrete structures, MMUPHin_Discrete a) evaluates the existence of discrete clusters within individual microbiome studies and b) validates the reproducibility of such structures among studies meta-analytically (Fig. 2e, f, Additional file 1: Fig. S9), by utilizing established clustering strength evaluation metrics [42]. For continuous structures, MMUPHin_Continuous identifies major axes of variation that explain the largest amount of heterogeneity among microbial profiles that are also consistent across studies. This is generalized from single study principal component analysis (PCA [43]) to multiple studies by constructing a network of correlated top PC loadings [35] (Fig. 2g, h, Additional file 1: Fig. S10). As a result, MMUPHin was able to successfully identify discrete clusters (i.e., microbiome "types") when present, as well as significantly consistent continuous patterns of microbiome variation that recur among populations (Additional file 2: Supplemental Notes).

### Meta-analysis of the IBD microbiome

Given these validations of MMUPHin's accuracy in simulated data, we next applied it to the 10-study, 4,789-sample IBD gut amplicon profile meta-analysis introduced above (Fig. 3). MMUPHin_Correct successfully reduced the effects both of differences among studies, and of batches within studies (study effect correction modelling disease and sample type as covariates, see "Methods"), although these remained among the strongest source of variation among taxonomic profiles as quantified by PERMANOVA R2 (Fig. 3a, "Methods," Additional file 5: Table S3). Among biological variables, sample type (biopsy/stool), biopsy location (multiple, conditional

on biopsy samples), disease status (IBD/control), and disease types (CD/UC, conditional on IBD) consistently had the strongest effect on the microbiome among studies. Several relationships between study design and phenotypic effects were apparent. Batches had a particularly strong effect in CS-PRISM and RISK, for example, where biopsy and stool samples were also perfectly separated by batch. Treatment exposures all had small effects on microbiome structure within studies, which typically reached statistical significance only when combined by meta-analysis; antibiotics were an exception with slightly larger effects. Montreal classification did not generally correspond with significant variation, while age (at sample collection as stratified below and above 18, and at diagnosis by Montreal age classification [44]) had small but significant effects. The effects of gender and race were not significant. Lastly, for longitudinal studies, relatively stable differences between subjects over time were large and significant, consistently for both longer-interval (HMP2) as well as densely sampled cohorts (Herfarth, daily samples), in agreement with previous individual studies' observations [9, 23].

We identified individual taxonomic features consistently associated with disease and treatment variables (Fig. 3b, Additional file 6: Table S4), with meta-analysis multivariate differential abundance analysis (MMUPHin_MetaDA), adjusting for common demographics (age, gender, race) and further stratifying for sample type and disease when appropriate ("Methods," Additional file 5: Table S3). At a very high level, differential abundance patterns between CD and control microbiomes were consistent with, and often more severe than contrasts between UC and control, confirming with increased resolution previous observations that CD patients tend to have more aggravated dysbiosis than UC patients [9]. As expected, our meta-analysis confirms many of the taxa associated with IBD reported by previous individual studies (Fig. 3b, detailed in Additional file 2: Supplemental Notes); they also agreed with important features as identified through other types of predictive, rather than hypothesis testing, machine learning models (Additional file 1: Fig. S11 [45]. These findings agree with the emerging hypotheses of pro-inflammatory aerotolerant clades (e.g., Escherichia and other Enterobacteriaceae) forming a positive feedback loop in the gut during inflammation, often of oral origin [7] (e.g., *Fusobacterium*, *Dialister*, *Veillonella*), and depleting the gut's typical fastidious anaerobe population as a result (primarily *Ruminococcaceae*, *Lachnospiraceae*, and other Clostridia and Firmicutes clades) [9].

We also identified two taxa not previously associated with IBD, both of modest effect sizes and likely newly detected by the meta-analysis' increased power. The genus *Acinetobacter* was enriched in CD, and *Turicibacter* was depleted. *Turicibater* in particular is poorly represented in reference sequence databases, with only nine genomes for one species (*Turicibacter sanguinis*) currently in the NCBI genome database; this makes it easy to overlook in shotgun metagenomic profiles relative to amplicon sequencing. The genus *Acinetobacter*, conversely, is quite well characterized due to its role in antimicrobial resistant infections [46], and it was previously linked specifically to the primary sclerosing cholangitis phenotype in UC [47], although without follow-up to our knowledge. *Turicibacter* is overall less characterized both in isolation and with respect to disease, although our findings and others' suggest it might be inflammation-sensitive when present; it was one of many clades increased in mice during CD8+ T cell depletion [48] and

reduced in a homozygous TNF deletion [49]. As the strains of *Acinetobacter* implicated in gut inflammation are unlikely to be those responsible for, e.g., nosocomial infections, further investigation of both clades using more detailed data or IBD-specific isolates is warranted.

Among treatment variables (samples or time points during which subjects were receiving antibiotics, immunosuppressants, steroids, and/or 5-ASAs), antibiotics had the strongest effects on individual taxa, as well as the greatest number of significantly associated taxa (Fig. 3b). These associations are also broadly in agreement with previous observations for microbiome responses to antibiotics in IBD or generally, e.g., the depletion of *Faecalibacterium*, *Ruminococcus*, and *Bacteroides* in patients treated with antibiotics, and the enrichment of (often stereotypically resistant) taxa such as *Streptococcus*, *Acinetobacter*, and the Enterobacteriaceae, with differential responses to the treatment groups speaking to both administration considerations and their impact on host versus microbial community bioactivities (Additional file 2: Supplemental Notes).

Subsets of IBD-linked taxa were additionally associated with the diseases' phenotypic severity (Fig. 4a, Additional file 7: Table S5). Montreal classification [44] was used as a proxy for disease severity, including Behavior categories for Crohn's disease (B1 non-stricturing, non-penetrating, B2 stricturing, non-penetrating, B3 stricturing and penetrating) and Extent for ulcerative colitis (E1 limited to rectum, E2 up to descending colon, E3 pancolitis). We tested for features differentially abundant in the more severe phenotypes when compared against the least severe category (B1 CD and E1 UC, "Methods"). Among statistically significant results, many extended those identified above as overall IBD associated (Fig. 3b), such as the depletion of *Faecalibacterium* in B3 CD and *Roseburia* in B2 CD, as well as the enrichment of Enterobacteriaceae in E3 UC. In most cases, microbial dysbiosis was also additionally aggravated from the moderate to the most extreme disease manifestations; such differences were statistically significant ("Methods") in, for example, the progressive depletion of *Bacteroides* in CD and UC, as well as the enrichment of Enterobacteriaceae in UC. This meta-analysis is uniquely powered to detect these subtle differences, which aid in shedding light on the microbiome's response to progressive inflammation and disease subtypes. Pancolitis corresponds with a unique microbial configuration distinct from regional colitis and not generally detectable in smaller studies [6], for example, while more severe CD induces essentially a more extreme form of the same dysbiosis observed in less severe forms of the disease.

Additionally, diseases (CD and UC) and their corresponding dysbioses also interacted distinctly with the microbiome under different treatment regimes and in different biogeographical environments (mucosa vs. stool, Fig. 4b, Additional file 8: Table S6). Interaction effects, in the statistical sense, were defined as a main exposure (IBD or treatment) having differential effects on taxon abundance with respect to either sample type (biopsy/stool) or diseases (CD/UC); they were identified via moderator meta-analysis models ("Methods"). Overall, we found elevated effects of both CD (relative to controls) and antibiotic treatment in stool as compared to biopsy-based measurements of the microbiome (Additional file 8: Table S6). An example of this is *Dehalobacterium*, with significantly greater depletion in CD stool relative to biopsies (Fig. 4b). *Dehalobacterium*, as with *Turicibacter* above, is underrepresented in reference sequence databases, better-detected by amplicon sequencing, and thus not a common microbial signature

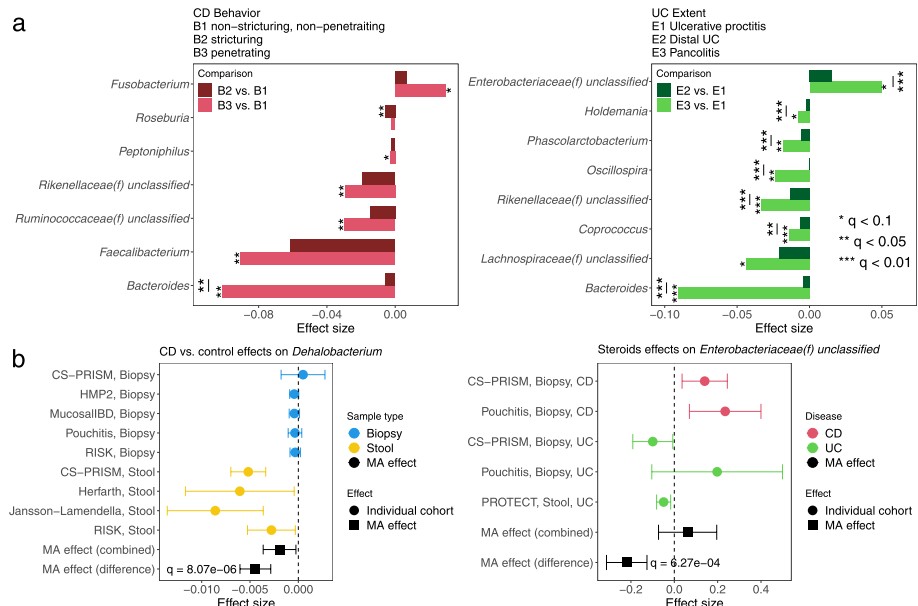

**Fig. 4** IBD-associated taxa are aggravated in more severe disease; disease biogeography and CD/UC differentially affect some taxa with respect to disease and treatment. **a** Statistically significant genera from meta-analytically synthesized differential abundance effects among severity of CD and UC phenotypes as quantified by Montreal classification. The difference between the most severe phenotype with the least severe one (B3 vs. B1 for CD, E3 vs. E1 for UC) was in most cases more aggravated than that of the intermediate phenotype. Many of the identified features overlap with those associated with IBD vs. control differences, suggesting a consistent gradient of severity effects on the microbiome. Individual study results in Additional file 7: Table S5. **b** Genus *Dehalobacterium* as an example in which a taxon is uniquely affected in the stool microbiome during CD and not at the mucosa. Likewise, family Enterobacteriaceae as an example in which steroid treatment corresponds with enrichment of the clade in CD samples, but depletion in UC. In all panels, effect sizes are aggregated regression coefficients on arcsin square root-transformed relative abundances. Full sets of statistically significant interactions, with individual study results, are in Additional file 8: Table S6

of IBD. It has been linked to CD in at least one existing 16S-based stool study [50]. In contrast, several UC-specific microbial disruptions were more prominent at the mucosa (i.e., in biopsies, Additional file 8: Table S6). Coupled with the severity-linked differences above, this suggests CD-induced changes in the entire gut microbial ecosystem largely as a consequence of inflammation, with UC-induced dysbioses both more local and more specific to disease and treatment regime. Additional results include effect of steroids on the Enterobacteriaceae, which tended to be more abundant in CD patients receiving steroids, but less abundant in UC recipients (Fig. 4b, Additional file 8: Table S6, Additional file 2: Supplemental Notes).

Lastly, we also conducted a more direct comparison of IBD microbiome associations found after applying each of the three batch correction methods (quantile normalization, ComBat, and MMUPHin_Correct) to our meta-analysis dataset. This employed a simpler post-correction testing strategy, as previously recommended [18], thus making the results more directly comparable but likely less biologically relevant than those discussed previously (Fig. 3). MMUPHin_Correct-processed abundance profiles still identified more IBD-associated genera compared to ComBat and quantile normalization, while also showing the best agreement with both other methods (Additional file 1: Fig.

S12). This provides empirical evidence for MMUPHin's effectiveness in real-world settings, in addition to its accuracy as quantified by our simulation studies.

### Consistent IBD microbial population structure discovered by unsupervised analysis

The existence of subtypes within gut microbial communities has been a major open question in human microbiome studies, and it is of particular importance within IBD as a potential explanation for heterogeneity in disease etiology and treatment response [6, 9]. To systematically characterize population structure in the IBD gut microbiome that was reproducible among studies, we performed both discrete (MMUPHin_Discrete) and continuous (MMUPHin_Continuous) structure discovery on the 10 cohorts using our meta-analysis framework. To identify potential discrete community types (i.e., clusters), we performed clustering analysis within each cohort's IBD patient population and evaluated the clustering strength via prediction strength (MMUPHin_Discrete, "Methods"). We found no evidence to support discrete clustering structure within individual cohorts, nor were we able to reproduce each cohort's clustering results externally (Fig. 5a). This lack of discrete structure was consistent when we further stratified samples to either CD or UC populations (Additional file 1: Fig. S13), or extended to additional dissimilarity metric and clustering strength measurements (Additional file 1: Fig. S13, "Methods"). Our observation that the IBD gut microbiome cannot be well characterized by discrete clusters is thus consistent with previous findings on gut microbial heterogeneity for healthy populations [41] and suggests that, at the level powered by this study, such microbiome subtypes are not clearly responsible for clinical heterogeneity.

Conversely, we identified two consistent, continuously varying gradients of microbial community variation in the IBD microbiome (Fig. 5b–d, Additional file 1: Fig. S14). These gradients represent patterns of microbes that occur with greater or lesser abundance in tandem, and which covary across subjects in a population; they were identified as principal component (PC) vectors that recur among different cohorts (MMUPHin_Continuous, see "Methods") [35]. Briefly, we used the four largest IBD cohorts (CS-PRISM, Pouchitis, PROTECT, and RISK) as training datasets to identify two clusters of consistent PCs (Fig. 5b), which were confirmed with sensitivity analysis (Additional file 1: Fig. S15) and validated in the remaining cohorts (Additional file 1: Fig. S16). The consensus loadings (i.e., within-cluster average) representing these two clusters (Fig. 5c, Additional file 1: Fig. S14, Additional file 9: Table S7) were used to assign continuously varying scores to the IBD population that capture gradient changes in the microbiome that occurred consistently within IBD, across diseases, sample types, and cohorts. This disease-linked "type" of microbiome variation corresponded roughly to severity or extent of inflammation, as detailed below.

In particular, while the second continuous population structure captured the Firmicutes-Bacteroidetes tradeoff present in most gut microbiome studies (Additional file 1: Fig. S14) [9, 32, 41], the first continuous score was IBD-specific and corresponded roughly to more extreme disease-associated dysbiosis in CD and UC populations (Fig. 5d). This is evidenced by the taxa with highest weights in the scores' consensus loading vector (Fig. 5c), which included taxa differentially abundant between IBD and control populations (Fig. 3). The score was consistent both within CD and UC while also further differentiating IBD, non-IBD control, and healthy populations (Fig. 5d,

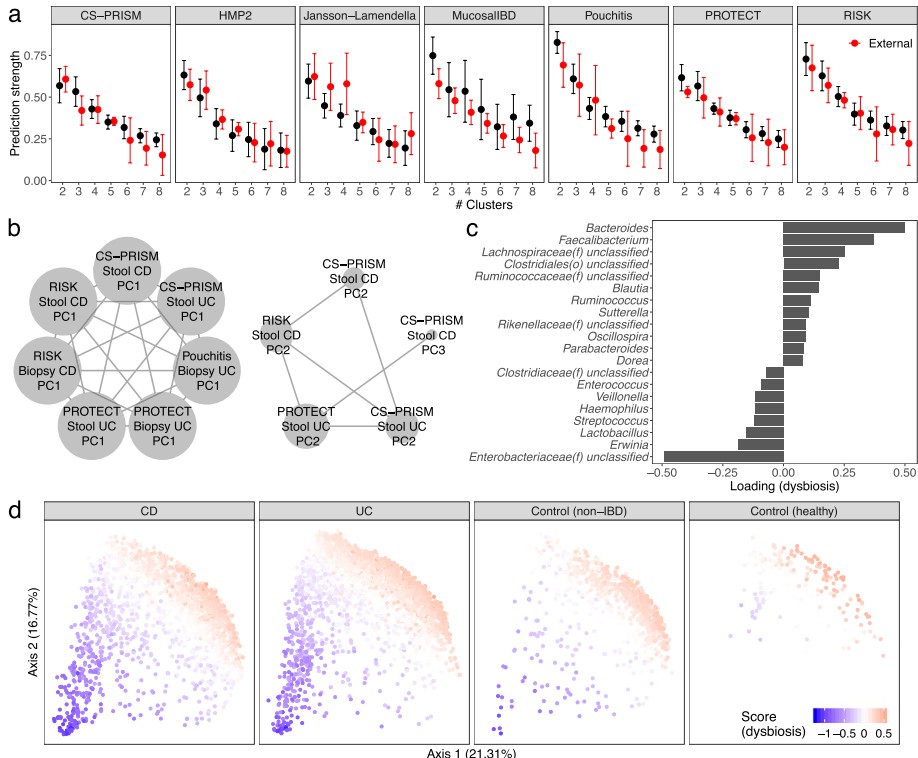

**Fig. 5** Unsupervised population structure discovery finds no evidence of microbiome-based subtypes in the IBD gut, but a reproducible gradient of continuously variable dysbiosis in disease. **a** No support was detected for discrete microbiome subtypes (clusters) within the IBD microbiome, neither within cohort nor when evaluated among studies (red bars) using prediction strength [42]. This remained true during stratification within CD and UC, and for additional dissimilarity metric/clustering strength measurements (Additional file 1: Fig. S12). **b** Conversely, two reproducible, continuously variable patterns of microbiome population structure were identified using groups of similar principal components ("Methods") [35]. These patterns were consistent within and between cohorts, disease types, and sample types, as well as under different edge strength cutoffs (Additional file 1: Fig. S14), and their consensus loadings were reproducible among cohorts (Additional file 1: Fig. S15). **c** Top 20 genera with highest absolute loadings for the disease-associated dysbiosis score corresponding to the first cluster in **b**. Many of these taxa were also IBD-associated (Fig. 3b). **d** Distribution of the dysbiosis pattern across CD, UC, non-IBD control, and healthy populations. Although it was defined in an unsupervised way solely within the IBD population, across which the pattern is highly variable, it also differentiates well between IBD and control populations (Additional file 1: Fig. S16)

Additional file 1: Fig. S17), even though it was identified unsupervisedly only from diseased subsets. The composition of the score and its population structure are also consistent with our recent definition of dysbiotic gut microbiome configurations corresponding with multi-'omic perturbations during IBD activity [9]. Together with the supervised meta-analysis results above, these unsupervised population structure findings confirm that there are no detectable discrete subtypes of the gut microbiome in IBD even among ~5000 combined samples, while showing a single continuously variable gradient of microbiome changes reproducibly present during more dysbiotic diseases.

## Discussion

Microbiome studies in general, and the IBD gut microbiome in particular, stand to benefit from meta-analysis, as have other multiply sampled conditions such as colorectal cancer [51, 52]. This is a major hurdle in establishing reproducibility of clinically relevant disease biomarkers, i.e., ecological and microbiological changes during disease that are reproducible across populations. Both the findings and methodology presented in this work are thus of interest, allowing MMUPHin to be validated and to leverage increased sample sizes from combined studies. However, all microbial community meta-analyses should be approached with caution, since in many cases unwanted sources of technical variation between studies (i.e., batch effects) are so large as to potentially mask biological signals even after correction [53–55] (Additional file 2: Supplemental Notes). Reducing interstudy variation in microbial community profiles is challenging relative to other 'omics data types due to (1) the extreme heterogeneity of microbes within most communities (exacerbating both technical and biological differences), and (2) feature zero inflation arising from both biological and technical reasons [13, 56]. MMUPHin alleviates this problem by taking care to incorporate batch/study effects in each of its components, not just during batch effect normalization (MMUPHin_Correct): mixed effects modelling was adopted for differential abundance meta-analysis to allow for residual per-study effects (MMUPHin_MetaDA); PC network clustering prioritizes consistent biological signals over batch effects (MMUPHin_Continuous). Thus despite these challenges, MMUPHin was able to meta-analyze amplicon profiles in this study both to associate microbial shifts with disease outcome, to associate them with treatment-specific differences, and to identify a single pattern of typical microbial variation within IBD. While previous efforts have developed IBD dysbiosis scores by contrasting patients with control groups [7, 9], this pattern of microbial variation was present specifically within IBD patients (both CD and UC), and in agreement with supervised methods, captured several classes of microbial functional responses in the gut (Additional file 2: Supplemental Notes).

We consider this study based on 16S rRNA gene sequencing to be a proof of concept, able to achieve unprecedented power due to the number of amplicon profiled samples available, but with greater precision possible in future work using, e.g., metagenomic and other 'omics technologies. This also enabled comparison of responses in the stool versus mucosal microbiomes, the latter of which are not amenable to metagenomic profiling from biopsies; these were in overall good agreement, but the few areas of significantly differential responses to inflammation are likely of particular immunological interest. The large sample and population sizes also provide some confidence in ruling out discrete, microbially driven population subtypes as an explanation for CD and UCs' clinical heterogeneity. Instead, the work identified a single consistent axis of gradient microbial change corresponding to increasing departures from "normal" microbiome configurations [7, 9, 57]. This pattern of consistent microbial dysbiosis can continue to be explored in further work on its functional, immunological, and clinical consequences. Overall, this study represents one of the first large-scale, methodologically appropriate, targeted meta-analysis of the IBD microbiome, and the corresponding methodology and its implementation are freely available for future meta-analyses of human-associated and environmental microbial populations.

## Conclusions

We provide a novel framework for microbial community meta-analysis and apply it to the first large-scale integration of over 5100 amplicon profiles of the stool and mucosal microbiomes in IBD. This identified a significantly reproducible gradient in the gut microbiome indicative of increasing dysbiosis in subsets of patients. Our results supported many of the taxonomic associations previously ascribed to IBD (e.g., *Faecalibacterium*, *Ruminococcus*, Enterobacteriaceae) while uncovering new associations (*Turicibacter*, *Acinetobacter*) not confidently associated with inflammation by other populations or data types. Almost all effects were exhibited similarly using either stool or mucosal profiling, with a small number of exceptions showing significant differentiation (e.g., *Dehalobacterium*). Novel disease-treatment response interactions were observed (e.g., steroids on Enterobacteriaceae). The study also showed no evidence of additional population structure, such as microbiome-driven discrete disease subtypes, within CD or UC. The meta-analysis framework developed for the study, MMUPHin, has been extensively evaluated and its performance for batch effect removal, supervised meta-analysis of exposures and covariates, and unsupervised population structure discovery validated on a variety of simulated microbial community types. It is extensible to integration of microbial community taxonomic or functional profiles from other data types (e.g., metagenomic sequencing), environments, or health conditions.

## Methods

### MMUPHin: a uniform statistical framework for meta-analysis of microbial community studies

We developed MMUPHin (Meta-analysis Methods with a Uniform Pipeline for Heterogeneity in microbiome studies) as a framework for meta-analysis of microbial community studies using taxonomic, functional, or other abundance profiles. It includes components for batch effect adjustment, differential abundance testing, and unsupervised discrete and continuous population structure discovery.

#### Batch adjustment: MMUPHin_Correct

For microbial community batch correction, we extended the batch correction method developed for gene expression data in ComBat [15] with an additional component to allow for the zero-inflated nature of microbial abundance data. In our model, sample read count $Y$ was modelled with respect to both batch variable and biologically relevant covariate(s) $X$:

$$Y_{ijp} = \exp\{\beta_p X_{ij}{}' + \sigma_p \left(\gamma_{ip} + \delta_{ip}\epsilon_{ijp}\right)\} \times I_{ijp}$$

where $i$ indicates batch/study, $j$ indicates sample, and $p$ indicates feature. $\gamma_{ip}$ and $\delta_{ip}$ are batch-specific location and scale parameters. $\sigma_p$ is a feature-specific standardization factor. $\beta_p$ are covariate-specific coefficients, and $\epsilon_{ijp}$ is an independent error term following a standard normal distribution. $I_{ijp}$ is a binary (0, 1) zero-count indicator, to allow for zero inflation of features. As in ComBat, $\gamma_{ip}$ and $\delta_{ip}$ are modelled with normal and inverse-gamma priors, respectively. Hyperparameters are estimated with empirical

Bayes estimators as in ComBat [15]. The posterior means, $\gamma\hat{*}_{ip}$ and $\delta\hat{*}_{ip}$, along with standard frequentist estimates $\hat{\beta}_p$ and $\hat{\sigma}_p$ are used to provide batch-corrected count data:

$$\tilde{Y}_{ijp} = \exp\left\{ \frac{Y_{ijp} - \hat{\beta}_p X_{ij}' - \gamma\hat{*}_{ip}\hat{\sigma}_p}{\delta\hat{*}_{ip}} + \hat{\beta}_p X_{ij}' \right\} \times I_{ijp}$$

Per-sample feature counts are then re-normalized to keep sample read depth unchanged post-correction. In practice, the user provides sample microbial abundance table (*Y*), batch/study information, and optionally any other covariates *X* that are potentially confounded with batch but encode important biological information. MMUPHin outputs an adjusted profile $\tilde{Y}$ that is corrected for the effect of batches but retains the effects of *X* (if provided).

With this model specification, we expect MMUPHin_Correct to often reduce, rather than fully correct batch differences. This is due to two considerations. First, MMUPHin_Correct focuses on correcting non-zero abundance batch effects, and does not change features' presence/absence across batches. "Correcting" a feature's batch-specific presence to absence is inappropriate, as substantial non-zero read counts indicate biological presence rather than technical artifacts. Imputing non-zero abundance for batch-specific absence is technically challenging in our linear modelling framework, as the per-sample/feature noise $\epsilon_{ijp}$ cannot be reliably inferred for inflated zero values. Second, the empirical Bayes batch effect estimates $\hat{\gamma_{ip}}*$ and $\hat{\sigma_{ip}}*$ are shrunken from their frequentist counterparts, which provides regularization for high-dimensional parameters as in ComBat and avoids "overfitting" to batch differences in small sample sizes. MMUPHin_Correct's design is thus intentionally conservative, by correcting batch differences that can be confidently inferred, and maintaining those that are not (which thus also avoids eliminating non-batch, biological signal).

Lastly, we note that MMUPHin_Correct does not explicitly model any particular sources of batch effects, such as primers, extraction protocols, and amplicon regions for 16S rRNA sequenced profiles. However, it will nevertheless attempt to correct for variability caused by differences in these protocols, to the extent that they manifest as batch/study differences. As examples: if two studies adopted different extraction protocols, potential study differences will be captured with MMUPHin_Correct and normalized. In contrast, if samples within the same study were sequenced using different amplicon regions, and this difference in protocol was not flagged as a "batch" variable, MMUPHin_Correct will not register the potential differences.

### Meta-analysis differential abundance testing: MMUPHin_MetaDA

For meta-analytical differential abundance testing, after batch correction, MMUPHin_MetaDA first performs multivariate linear regression within individual studies using previously validated data transformation and modelling combinations appropriate for microbial community profiles (MaAsLin2 [33]). This yields study-specific, per-feature differential abundance effect estimations $\hat{\beta}_{ip}$, where *i* indicates study and *p* indicates feature. These are then aggregated into meta-analysis effect size with fixed/random effects modelling as implemented in the metafor R package [34]:

$$\hat{\beta}_{ip} = \beta_p + \epsilon_{ip} + e_{ip}$$

$\beta_p$ is the overall differential abundance effect of feature $p$. $\epsilon_{ip}$ is per-study measurement error, and $e_{ip}$ is study-specific random effects term (not present in fixed-effect models).

Overall, for running MMUPHin_MetaDA, the user provides a microbial community profile, study design (batch) information, the main exposure variable of interest, and optional additional covariates. If any meta-analyzed studies include repeated measures (e.g., longitudinal designs), then random covariates can also be provided and will be modelled for such studies. MMUPHin_MetaDA then performs MaAsLin2 regression modelling within each study and aggregates effect sizes of the exposure variable $\hat{\beta}_{ip}$ across studies using the resulting random/fixed effects model. The estimated overall effect, $\hat{\beta}_p$, is reported as the overall differential abundance effect for feature $p$.

We note that MMUPHin_MetaDA always accounts for the batch variable in its supervised differential abundance testing. This agrees with the field's consensus on the most appropriate way to address batch effects during supervised testing [15, 58]. Through simulation evaluations, the performance (FPR, power) of MMUPHin_MetaDA is robust with or without upstream adjustment with MMUPHin_Correct (Additional file 1: Fig. S8). Nevertheless, pre-correcting the data with MMUPHin_Correct can still be helpful. This is both consistent with similar applications of batch correction in other molecular data types [15], and because MMUPHin_Correct accounts for both location and scale batch effects, while the linear modeling in MMUPHin_MetaDA only accounts for the former. Regardless, correcting the data with MMUPHin_Correct is most useful in analysis tasks where accounting for batch effects is otherwise not straightforward, such as for visualizing the data or during unsupervised population structure discovery.

### Unsupervised discrete structure discovery: MMUPHin_Discrete

For unsupervised discrete (i.e., cluster) structure discovery of a single study, again after batch correction, MMUPHin_Discrete uses average prediction strength [42], an established clustering strength metric, to measure the existence of reproducible clusters among meta-analyzed datasets. Briefly, for each individual dataset, the metric randomly and iteratively divides samples into "training" and "validation" subsets. In each iteration, clustering is first performed on the training samples, across a range of cluster numbers $k$, yielding (for a specific $k$) training sample clusters $A_{k1}, A_{k2}, ..., A_{kk}$. Note that $A_{k1}, A_{k2}, ..., A_{kk}$ jointly forms a partition of the testing sample indices. The same clustering analysis is then performed on the validation samples, and the resulting partition of sample space provides classification membership potentially different from clustering memberships $A_{k1}, A_{k2}, ..., A_{kk}$. Prediction strength for $k$ clusters is defined as

$$ps(k) = \min_{1 \leq l \leq k} \frac{1}{n_{kl}(n_{kl} - 1)} \sum_{j \neq j' \in A_k l} I\{\text{validation samples } j \text{ and } j' \text{ are classified to the same group according to training samples}\}$$

i.e., the minimum (across validation clusters) proportion of same-cluster sample pairs also being classified as the same group by training samples. $n_{kl} = |A_{kl}|$, or the number of test samples in the $l$th cluster.

Average prediction strength is the average of prediction strengths across randomization iterations. Intuitively, it characterizes the degree of agreement between the

clustering structures in randomly partitioned validation and training subsets; if $k$ is appropriately describing the true number of discrete clusters in the dataset, then average prediction strength should be close to one (training and validation samples agree most of the time).

We additionally generalized this metric to meta-analysis settings, where we aimed to quantify the agreement of clustering structures between studies. In the meta-analytical setting, generalized prediction strength for cluster number $k$ in study $i$ with validation study $i'$ is

$$gps_{ii'}(k) = \min_{1 \le l \le k} \frac{1}{n_{kil}(n_{kil} - 1)} \sum_{j \ne j' \in A_{kil}} I\{\text{validation samples } i'j \text{ and } i'j' \text{ are classified to the same group according to study } i\}$$

where $A_{kil}$ indicates the $l$th cluster membership in study $i$, when cluster number is specified as $k$; $n_{kil} = |A_{kil}|$. The average generalized prediction in study $i$ for cluster number $k$ is then defined as the average of $gps_{ii'}(k)$ across all $i' \ne i$, i.e., all validation studies (instead of iterations of randomized partitions). Similar to the single study prediction strength, it describes the generalizability of clustering structure in study $i$ in external validation studies.

### Unsupervised continuous structure discovery: MMUPHin_Continuous

We extended our previous work in cancer gene expression subtyping [35] to perform unsupervised continuous structure discovery in microbial community profiles. Complementary to discrete cluster discovery, the goal is to identify strong feature covariation signals (gradients) that are reproducible across studies. This is carried out by performing principal component analysis individually in microbiome studies and constructing a network of correlated PCA loading vectors, to identify loadings that are consistently present across studies. In detail, given a collection of training microbial abundance datasets, our method takes the following steps (visualized in Additional file 1: Fig. S5):

(1) For each dataset $i$, PCA is performed on normalized and arcsin square root-transformed microbial abundance data. Given a user-specified threshold on variance explained, we record its top PC loading vectors, $w_{i1}, w_{i2}, \ldots, w_{iJ_i}$, where $J_i$ is the smallest number of top loading vectors that jointly explain percentage of variability in the dataset past a customizable threshold $0 < threshold_v < 1$ (default to 80%).

(2) For two PC loadings from different datasets $w_{ij}$ and $w_{i'j'}$, similarity is quantified with the absolute value of cosine coefficient [59] $|cos < w_{ij}, w_{i'j'}>|$. This yields a network of PC loading vectors associated by weighted edges $w_{ij}$ and $w_{i'j'}$, retaining edges only if their weight surpasses a customizable similarity threshold ($|cos < w_{ij}, w_{i'j'}>| > threshold_s, 0 < threshold_s < 1$).

   (a) This threshold is default to 0.7, which is close to the theoretical guarantee that all size-three clusters will by definition have positive cosine coefficients between all PC pairs. In practice, we recommend the user to vary this parameter as needed to evaluate robustness and interpretability.

(3) In the resulting network, we perform cluster detection based on modularity score [60, 61] to identify densely connected modules of PCs. Each module by definition

consists of PCs from different datasets that are similar to each other—whether or not they occur in the same order or with similar percent variance explained—and which thus represent strong feature covariation signals that are recurrent in studies.

(a) Clustering by modularity score avoids large clusters with few intracluster edges and prioritizes smaller clusters that are more densely connected (Additional file 1: Fig. S18, Additional file 2: Supplemental Notes). This is relevant for MMUPHin because the more densely connected a cluster is, the better consistency the PCs in the cluster have, which provides evidence for recurring biological signals across the spanned datasets.

(4) For a module $k$ containing PC set $M_k$, its consensus vector $W_k$ is calculated as the average of sign-corrected loading vectors in $M_k$, i.e., $W_k := \frac{\sum_{w_{ij} \in M_k} \widetilde{w_{ij}}}{|M_k|}$. Note that the average is taken not over the original loading vectors $w_{ij}$, but rather their sign-corrected versions $\widetilde{w_{ij}}$. Specifically, the signs of each $w_{ij}$ in $M_k$ are corrected so that all loading vectors have positive cosine coefficients.

(a) We note that, given a specific cosine threshold for constructing edges of the network, it is not guaranteed that such a correction is always possible. That is, with all possible sign corrections, there are still certain intracluster PCs that have negative cosine coefficients. Such cases are unlikely to happen in empirical evaluations and are further reduced by our modularity clustering approach (Additional file 1: Fig. S18). We discuss this issue in Additional file 2: Supplemental Notes.

(b) In the case where such issues occur, a higher cosine threshold is recommended. With a sufficiently high cosine threshold, clusters are guaranteed to be consistent (all PCs will have positive cosines), but also be smaller and thus are less interpretable in terms of consistent biological signals across studies.

(5) The module-wide consensus vectors $W_k$ represent strong, mutually independent, and reproducible covariation signals across the microbial datasets; they are used to identify continuously varying gradients in microbial abundance profiles that represent reproducible population structures. Specifically, given a sample with normalized and transformed microbial abundance measurements $x$, its continuous score for module $k$ is defined as $x' W_k$, as in regular PCA.

(6) If additional studies are available, the reproducibility of each $W_k$ can be further examined by correlating $W_k$ with the top PC loadings in each such validation study. For each additional study, $W_k$ is considered to be validated in that dataset if its absolute cosine coefficient with at least one of the dataset's top PCs surpasses the coefficient similarity cutoff $threshold_s$; the number of top PCs to consider in the validation dataset loadings is determined with the same cutoff $threshold_v$.

### Simulation validation of MMUPHin

We performed extensive simulation studies (Fig. 2, Additional file 1: Figs. S6-S10, Additional file 4: Table S2) to validate the performance of each component of MMUPHin

(MMUPHin_Correct, MMUPHin_MetaDA, MMUPHin_Discrete, and MMUPHin_Continuous). In all cases, these employed realistic microbial abundance profiles generated using SparseDOSSA (http://huttenhower.sph.harvard.edu/sparsedossa). This is a model of microbial community structure using a set of zero-inflated log-normal distributions fit to selected training data, in this case drawn from the IBD gut microbiome [6]. Controlled microbial associations with simulated covariates can then (optionally) be spiked in. Note that although the assumed null distributions in MMUPHin and SparseDOSSA are the same (zero-inflated log normal), the models of effects for batch and biological variables are substantially different: MMUPHin assumes exponentiated effects, while SparseDOSSA assumes re-standardized linear effects.

Specifically, SparseDOSSA models null microbial feature abundances using a zero-inflated log-normal distribution:

$$Y_{ip} \sim LogN\left(\mu_p, {\sigma^2}_p\right) \times Bernoulli\left(\pi_p\right)$$

This is the same initial distributional assumption as the MMUPHin batch correction model, when there are no batch or covariates effects. However, for spiked-in associations with metadata (batch, biological variables, etc.), SparseDOSSA uses a different model. Given a simulated, pre-spiking-in feature count vector $Y_p$ with mean $\mu_p^Y$ and standard error $\sigma_p^Y$, as well as a metadata variable vector $X$ with mean $\mu^X$ and standard error $\sigma^X$, the post-spiked-in feature count is set to:

$$\widetilde{Y_{ip}} = \frac{1}{1+\phi}\left\{Y_{ip} + \phi \times \left[\frac{(X_i - \mu^X)\sigma_p^Y}{\sigma^X} + \mu_p^Y\right]\right\}$$

where $\phi$ is a configurable spike-in strength parameter. By this definition, microbial features post-spike-in have the same mean and approximately the same variance as before, the only difference being the added association with the metadata variable(s) used. This is to ensure the counts of the modified feature are not dominated by the values of the target covariate, but instead distributed similarly to real data. The SparseDOSSA association model thus differs from MMUPHin's model in two substantial ways: (i) MMUPHin's associations are defined within the exponentiated component and are thus better described as a multiplicative effect, whereas SparseDOSSA's effects are directly applied on untransformed data, and (ii) SparseDOSSA additionally ensures realistic data generation with the re-standardization procedure.

Thus, the only component of the SparseDOSSA model that requires fitting to training data is the aforementioned zero-inflated log-normal null distribution. In our analysis, this was always PRISM [6], while other parameters were specified across a wide range of combinations to simulate different application scenarios. These include the effect sizes of the associated batch and biological variables (i.e., the $\phi$ parameter), number of batches, sample sizes, and dimensionality (both the total number of features and the percentage of features randomized to be associated with batch/biological variables). For each combination of simulation parameters, we performed 20 random replications (i.e., running simulation/evaluation with the same parameters but different random seeds). Additional file 4: Table S2 presents the full list of parameter combinations.

### Evaluating batch adjustment

For evaluation of MMUPHin's batch effect adjustment component, MMUPHin_Correct, we simulated metadata that included batch (with varying total batch numbers 2, 4, 6, 8), a binary positive control (simulated "biological" covariate), continuous positive control ("biological"), and negative control (binary, and guaranteed to be non-associated with microbial features) variables. Microbial abundance data was simulated to be associated with the batch and the two positive control variables at varying effect sizes (1, 2, 5, 10 for batch variable and fixed at 10 for positive control variables), but not with the negative control variable. We additionally varied the number of samples per batch (20 to simulate multiple-batches in a single study scenario, 100 to simulate meta-analysis with moderate sized studies and 500 to simulate large meta-analysis), total number of microbial features ($n$=200 and 1000), as well as the percentage of features associated with metadata (5%, 10%, and 20%) (Additional file 4: Table S2).

Performance of batch correction methods was quantified by omnibus associations (PERMANOVA R2) between the simulated microbial abundance data with the batch and positive control variables, before and after batch correction. For ComBat [15] and our method, batch correction was performed with both positive control variables and the negative control variable as covariates. MMUPHin_Correct successfully reduced the confounding batch effect, but retained the effect of positive control variables, and did not inflate the effect of negative control variable (Fig. 2a, Additional file 1: Fig. S6).

### Evaluating meta-analytic differential abundance testing

We evaluated false positive rates (FPR) for meta-analytic feature association testing, specifically the null case in which there are no associations between microbial features and covariates, but false associations can arise in the presence of batch effects with unbalanced distribution of covariate values across studies (Fig. 2b). For simulation, we generated a binary covariate unevenly distributed between two "studies" at varying levels of disparity (Additional file 4: Table S2). Microbial abundance data was simulated to be associated only with the two studies and not with the covariate (i.e., study confounded null data), with varying strengths of batch effect (from 0 to 10). The number of samples per batch varied between 100 and 500 to, again, simulate moderate- and large-sized meta-analysis. Lastly, we varied a total number of microbial features and the percentage of features associated with metadata as above.

FPRs were calculated as the percentage of simulated microbial features with nominal $p$-values < 0.05 for associations with the exposure variable. Four data normalization and analysis regimes were evaluated (Fig. 2c, Additional file 1: Fig. S6): (a) naive MaAsLin2 model on the study effect confounded null data (without explicitly modelling the batches), (b) the quantile normalization procedure, paired with two-tailed Wilcoxon tests, as proposed in [18], (c) BDMMA as proposed in [19], with the default 10,000 total MCMC sampling and 5000 burn-in, (d) the complete MMUPHin meta-analysis model for the batch-corrected data as described above (MMUPHin_Correct + MMUPHin_MetaDA). Note that due to its computational cost we were only able to evaluate the Dirichlet-multinomial regression model on a subset of parameter combinations, namely number of samples per batch = 100, number of features = 200, and percent of associated

microbes = 5%. These parameters roughly agree with those used in the simulation analysis in the method's original publication [19].

We also evaluated the computational costs of quantile normalization, BDMMA, and MMUPHin (Additional file 1: Fig. S7). For this, the same subset of 20 replications (batch effect 0, exposure imbalance 0, number of samples per batch 100, and number of features 200) were ran through the three methods under the same computation environment (single core Intel(R) Xeon(R) CPU E5-2680 v2 @ 2.80GHz). The computational cost of BDMMA is prohibitive when compared to MMUPHin and quantile normalization, requiring ~5 total CPU hours to finish on the very moderately sized data (200 total samples by 200 features).

### *Evaluating unsupervised discrete structure discovery*

To simulate microbial abundance data with known discrete clustering structure, we again used the simulation model above, with microbial feature associations added both with a discrete "batch" variable and a discrete clustering variable, at varying number of batches (2, 4, 6, 8), number of clusters (3, 4, 5, 6), and effect size of association (0 to 10 for batch, fixed at 10 for cluster). For the evaluation of MMUPHin's unsupervised methods (both here for MMUPHin_Discrete and during continuous population structure discovery below for MMUPHin_Continuous), we fixed the number of samples per batch at 500, the number of total features at 1000, and the percent of associated features at 20%. These were guided by the fact that the underlying unsupervised methods (clustering, PCA) require larger sample sizes for good performance even without batch confounding, and are generally only practical with higher feature dimensions (Additional file 4: Table S2).

Performance of clustering was evaluated as the percentage of replicates in which the right number of synthetically defined underlying clusters was identified using prediction strength, across technical replicates for a fixed combination of simulation parameters. That is, the number of clusters within a simulation was identified as that which maximized prediction strength. This was compared to the "truth" (i.e., the known simulation parameter) and counted as a success only if the two agreed. The percentage of success for a given parameter combination across the 20 random replications was used as the evaluation metric for model performance. We compared the performance of clustering before and after MMUPHin_Correct (Fig. 2e, Additional file 9: Table S7). Note that batch correction is modelled only using the batch variable and specifically not including the cluster variable as a covariate in the batch correction model above, as the underlying cluster structure is unknown in non-synthetic unsupervised analyses settings.

### *Evaluating unsupervised continuous structure discovery*

To simulate microbial abundance data with known continuously variable population structure, we spiked in feature associations with both a simulated batch covariate (4, 6, 8) and a continuously varying gradient (uniformly distributed between −1 and 1), at varying number of batches and effect size of both associations (as above). The number of samples per batch, total number of microbial features, and the percentage of features associated were fixed at the same values as above (Additional file 4: Table S2).

Performance of continuous structure discovery analysis with MMUPHin_Continuous was evaluated as the Spearman correlation between the known simulated gradient score and the strongest continuously valued population structure as identified by MMUPHin's continuous structure discovery method (above). We again compared the performance of continuous score discovery on the batch confounded and batch-corrected data (Fig. 2g, Additional file 1: Fig. S10). Note that, as above, batch correction is again modelled only using the batch variable and does not have any access to the synthetic continuous gradient, as any underlying continuous population structure is unknown during unsupervised analyses settings.

### Collection and uniform processing of ten IBD microbiome studies employing 16S rRNA gene sequencing

#### *Study inclusion and raw sequence data*

We curated 10 published 16S rRNA gene sequencing (abbreviated 16S) gut microbiome studies of IBD for meta-analysis (Table 1, Additional file 3: Table S1). Demultiplexed raw sequences were either downloaded from EBI (Jansson-Lamendella and Herfarth) or available locally as previously generated (other eight studies). Metadata were obtained either directly from the sequence repository/manuscript (Herfarth, Jasson-Lamendella, HMP2, MucosalIBD, PROTECT, RISK), or from collaborators (BIDMC-FMT, CS-PRISM, LSS-PRISM, Pouchitis). This resulted in a total of 5151 samples and 2179 subjects available prior to processing and quality control.

#### *Metadata curation*

We manually curated subject- and sample-specific metadata across studies to ensure consistency. Variables collected and curated include:

- Disease (CD, UC, control), universally available.
- Type of controls (non-IBD, healthy). Control information was available directly for CS-PRISM, Jansson-Lamendella, and Pouchitis, inferred from study design described in manuscript for Herfarth, HMP2, MucosalIBD, and RISK (all non-IBD controls), and not applicable for BIDMC-FMT, LSS-PRISM, and PROTECT (only has IBD subjects).
- Sample type (biopsy, stool), universally available.
- Body site of biopsy sample collection (ileum, colon, rectum), with more detailed classifications recorded separately in case of need. Mappings for the relevant datasets are:

    ° CS-PRISM: terminal ileum, neo-ileum, pouch are aggregated as ileum; cecum, ascending/left-sided colon, transverse colon, descending/right-sided colon, and sigmoid colon were aggregated as colon; rectum classification was kept unchanged.

° HMP2: ileum classification kept unchanged; cecum, ascending/right-sided colon, transverse colon, descending/left-sided colon, and sigmoid colon were aggregated as colon.

° MucosalIBD: all terminal ileum samples, aggregated to ileum.

° Pouchitis: terminal ileum, pouch, pre-pouch ileum aggregated as ileum; sigmoid colon aggregated to colon.

° PROTECT: all rectum samples, classification kept unchanged.

° RISK: terminal ileum was aggregated to ileum; rectum kept unchanged.

- Montreal classifications:

  ° Location for CDs (L1, L2, L3, and possible combinations), available for BIDMC-FMT, CS-PRISM, Herfarth, Jansson-Lamendella, LSS-PRISM, and Pouchitis.

  ° Behavior for CDs (B1, B2, and B3), available for CS-PRISM, Herfath, Jansson-Lamendella, LSS-PRISM, Pouchitis, and RISK.

  ° Extent for UCs (E1, E2, and E3), available for CS-PRISM, Jansson-Lamendella, LSS-PRISM, Pouchitis, and PROTECT.

- Age at sample collection (in years), available for BIDMC-FMT, CS-PRISM, Herfarth, HMP2, LSS-PRISM, MucosalIBD, Pouchitis, PROTECT, RISK.
- Age at diagnosis (in years). Directly available for CS-PRISM, HMP2, LSS-PRISM, and Pouchitis, inferred as baseline age for PROTECT and RISK as these were new-onset cohorts.
- Race (White, African American, Asian / Pacific Islander, Native American, more than one race, others). Directly available for CS-PRISM, Herfarth, HMP2, PROTECT, and RISK, inferred from manuscript cohort description for Jansson-Lamendella (all Caucasian cohort).
- Gender (male/female). Available for BIDMC-FMT, CS-PRISM, Herfarth, HMP2, Jansson-Lamendella, LSS-PRISM, MucosalIBD, Pouchitis, PROTECT,
- Treatment variables, including antibiotics, immunosuppressants, steroids, and 5-ASA. These variables were encoded as yes/no to indicate, approximately, currently receiving them at the time of sampling. Additional information such as specific medication or delivery method was recorded separately if available in case of need. We note the potentially confounding difference in studies' definitions of treatment: for Pouchitis and PROTECT authors defined antibiotics as receiving the treatment within the past month (30 days for Pouchitis, 27 days for PROTECT), whereas for CS-PRISM, HMP2, LSS-PRISM, and RISK such determination was not possible (antibiotics "yes" was defined as "currently taking"). Likewise, we had no additional information to determine the time extent for the other three treatments, beyond that according to metadata/publication, patients were "currently taking" the treatment at sample collection.

For a comprehensive list of curation mapping schema, please refer to our metadata curation repository: https://github.com/biobakery/ibd_meta_analysis.

### *16S amplicon sequence bioinformatics and taxonomic profiling*

Sequences were processed, per cohort, with the published, standardized bioBakery workflow [62] using the UPARSE protocol [63] (version v9.0.2132-64bit). For all studies, demultiplexed sequences were truncated at 200bp max length and filtered by maximum expected error of one [63]. Operational taxonomic units (OTUs) were clustered at 97% identity and aligned using USEARCH with 97% identity to the Greengenes database 97% reference OTUs (version 13.8) [64] for taxonomy assignment. The resulting Greengenes identifiers for OTUs were used as basis for matching features (taxa) among cohorts.

### *Quality control*

Across samples, a median of 81.51% reads / sample passed quality control filtering and were successfully assigned to OTUs with Greengenes identifiers (Additional file 1: Fig. S1). These 8921 raw OTUs aggregated to a total of 1122 genera prior to quality control. We retained taxa that exceeded 5e−5 relative abundance with at least 10% prevalent in at least one study; this criterion generally removes spurious OTU assignments while retaining rare organisms if confidently present in at least one study. Lastly, we also removed low read depth samples with less than 3000 total sequences, which retained 78.34–100% samples per cohorts (Additional file 3: Table S1). The final resulting taxonomic profile, used for all further analysis, aggregated into 249 total genera spanning 4789 samples (OTUs unclassified under a particular taxonomy level were aggregated as "unclassified" feature under that taxon, e.g., "Enterbacteriaceae unclassified" accumulates all OTUs' abundances under the family that could not be classified at the genus level.

### *Data availability*

Quality-controlled (truncated and filtered) sequences, Greengenes mapped OTU count profiles, and curated sample metadata are available at the Human Microbial Bioactives Resource Portal (http://portal.microbiome-bioactives.org).

### Applying MMUPHin to IBD gut microbiome meta-analysis

For the resulting collection of microbiome studies, batch and study effects were performed using MMUPHin_Correct on the genus level feature abundance profiles. Batch (i.e., sequencing run) effect correction was first performed within individual studies (when batch/plate information was available, applicable to BIDMC-FMT, CS-PRISM, LSS-PRISM, MucosalIBD, and RISK). Microbial abundance profiles across all studies were then jointly corrected for study effects, while modelling disease status (IBD or control), disease (CD or UC), and sample type (biopsy or stool) as covariates. Reduction of batch and study effects was evaluated by PERMANOVA R2 (Fig. 3a).

### *Sensitivity analysis for amplicon sequence variants (ASV) microbial abundance profiles*

We additionally evaluated the impact of OTU- versus ASV-based bioinformatics pipelines on our method and meta-analysis results (Additional file 1: Fig. S18). Specifically, we processed two studies representing extremes of size, BIDMC-FMT ($n=16$, two technical batches) and RISK ($n=882$, fifteen batches) with the dada2 method [65], and performed batch correction and evaluation on the generated ASV profiles. MMUPHin_Correct was

still capable of reducing batch effects in either study's ASV-based abundance profiles, showcasing that our method is applicable to such new bioinformatics protocols (Additional file 1: Fig. S18a). Additionally, when aggregated at the genus level, OTU- and ASV-based abundance profiles had limited differences, suggesting that the choice of sequence variant units has limited impact on our meta-analysis results, as previously indicated [66] (Additional file 1: Fig. S18b).

### Association analyses

#### *Omnibus testing of microbial composition associations*

We used PERMANOVA tests (2000 permutations) as implemented in the R package vegan [38] using Bray-Curtis dissimilarities for all omnibus association tests of overall microbial community structure with covariates (Fig. 3a). Where appropriate, R2s were calculated conditioning on the necessary covariates; specifically, CD/UC Montreal classifications were conditional on CD/UC samples respectively, treatment was conditional on IBD status, biopsy location was conditional on a sample being a biopsy, and all covariates were conditional on being non-missing. Otherwise, variables were tested marginally (that is, each as the sole variable in the model). Importantly, to account for repeated measures within subjects for longitudinal studies, we adopted the blocked permutation strategy as in [9], where per-sample measurements (sample type, biopsy location, treatment) were permuted within subjects, and per-subject measurements (disease, demographics) were permuted along with subjects (but within cohorts, relevant for the all-cohort evaluation). For a full list of the model and permutation strategies that this resulted in for our analysis, please refer to Additional file 5: Table S3. Finally, per-variable *p*-values were adjusted with Benjamini-Hochberg false discovery rate control on a per-study basis.

#### *Per-feature meta-analysis differential abundance testing*

To identify microbial features individually significantly associated with one or more covariates, we applied MMUPHin's differential abundance testing model (MMUPHin_MetaDA) as described above. Cohorts were first stratified by sample type (biopsy or stool) and, where appropriate, diseases (CD or UC) prior to model fitting. Arcsin square root-transformed genus level taxon abundances were tested for covariate associations in individual cohort strata with multivariate linear modelling (linear random intercept model adopted for longitudinal studies). Covariates used for adjustment include age, gender, and race for disease variables, and additionally disease status for treatment variables. Effect sizes across cohort strata were aggregated with a random effects model with restricted maximum likelihood estimation [34]. *P*-values were FDR adjusted across features for each variable. For the full list of models adopted as well as cohort stratification strategy, please refer to Additional file 5: Table S3. Figure 3b visualizes the aggregated meta-analysis effects; for individual study results, refer to Additional file 6: Table S4.

To relate these real data results to our simulation evaluation, the real-world data characteristics best correspond with the simulated scenario with eight batches and four thousand samples in total (versus ten real studies and 4789 samples post filtering), 200 microbial features (249 real genera), and 10% spiked features at batch effect size 10, which yielded ~10% PERMANOVA R2 for batch effect and 3% R2 for binary exposure

(10.98% for studies and 3.48% for sample type observed in real data) (Additional file 1: Fig. S6 panels, second row, last column).

### Testing for phenotypic severity within CD and UC patients

Meta-analytical testing of features associated with CD behavior and UC extent classifications were performed with similar models (Additional file 5: Table S3). Specifically, within each study's CD patients, the tests for contrasts B2 versus B1 and B3 versus B1 are performed by

$$\text{Relative abundance} \sim \beta_0 + \beta_1 I\{\text{subject is B2}\} + \text{additional covariates (subsetted to B1, B2 CDs)}$$
$$\text{Relative abundance} \sim \beta_0 + \beta_1 I\{\text{subject is B3}\} + \text{additional covariates (subsetted to B1, B3 CDs)}$$

The two $\beta_1$ coefficients, once aggregated with meta-analysis, were reported as the effect sizes shown in Fig. 4a, along with their FDR corrected $q$-values (adjusted across features for each test).

$$\text{Relative abundance} \sim \beta_0 + \beta_1 I\{\text{subject is } B2 \text{ or } B3\} + \beta_2 I\{\text{subject is } B3\} + \text{additional covariates}$$

$\beta_2$ in this model corresponds to the effect of B3 in addition to the overall contrasts between B23 versus B1. The meta-analysis aggregated $p$-values of these effects were reported as the differentiation between the most severe and "medium" severity phenotypes (vertical bars indicating significance in Fig. 4a). Note that FDR adjustment of this effect was performed across the subset of features with at least either B2 versus B1 or B3 versus B1 effect significant (i.e., the subset of features visualized in Fig. 4a). Equivalent models were adopted for contrasts between extent categories of UC patients. Individual study results for the aggregated effects in Fig. 4a are in Additional file 7: Table S5.

### Interaction effects testing

To test for interaction effects with sample type and diseases, we fit meta-analysis moderator models [34] on the per cohort strata effects:

$$\hat{\beta}_{ip} = \beta_{0p} + \beta_{1p} I\{\text{cohort strata } i \text{ is biopsy}\} + \epsilon_{ip} + e_{ip}$$
$$\hat{\beta}_{ip} = \beta_{0p} + \beta_{1p} I\{\text{cohort strata } i \text{ is CD}\} + \epsilon_{ip} + e_{ip}$$

The moderator effects $\beta_{1p}$ correspond to the interaction effect between the exposure under evaluation (disease, treatment, etc.) with the moderator variable. Figure 4b visualizes the two example features, *Dehalobacterium* and Enterobacteriacea; all significant interactions as well as individual study effects are in Additional file 8: Table S6.

## Population structure analyses

### Discrete structure discovery

We performed discrete subtype discovery (i.e., "enterotyping" [67]) in IBD, CD, and UC populations across studies (longitudinal studies subsetted to baseline samples), using MMUPHin's discrete structure discovery component (MMUPHin_Discrete). Only studies with at least 33 samples were considered for clustering analysis, as this was the sample size in the original enterotype paper [32]. Specifically, clustering was performed on Bray-Curtis dissimilarity by the partition-around-medoid method as implemented in R

package cluster; the same method was adopted in previous enterotyping efforts including the original enterotype paper [32, 41]. Clustering was evaluated with prediction strength and validated externally with MMUPHin's generalized prediction strength as described above. Across studies, we found no evidence to support a particular number of clusters within IBD, CD, or UC populations (Fig. 5a, Additional file 1: Fig. S12), suggesting that the IBD microbiome does not have discrete clusters.

We additionally extended our clustering evaluation analysis to other dissimilarity metrics (Jaccard, root Jensen-Shannon divergence) and clustering strength measurements (Calinski-Harabasz index, average silhouette width), which were also explored in previous efforts [41]. Importantly, the original enterotype paper adopted root Jensen-Shannon divergence and Calinski-Harabasz index for cluster discovery. Across combinations of these additional dissimilarities and clustering strength metrics, we also found no evidence to support discrete clusters (Additional file 1: Fig. S12).

### Continuous structure discovery

Continuous structure discovery was performed with MMUPHin's corresponding component (MMUPHin_Continuous). The four largest studies (CS-PRISM, Pouchitis, PROTECT, RISK) were subsetted to baseline samples (only relevant for PROTECT), stratified by CD/UC and biopsy/stool sample type, and used as the training sets for MMUPHin_Continuous. The minimum variance explained threshold ($\text{threshold}_v$) was set to default (80%), but we varied the PC similarity (evaluated by absolute cosine coefficient) cutoff $\text{threshold}_s$ between 0.5 and 0.8 to assess the sensitivity of the two identified PC clusters in Fig. 5b (corresponding to $\text{threshold}_s = 0.65$). As we show in Additional file 1: Fig. S14, with a small $\text{threshold}_s$(0.5) PC networks become denser, with the two PC clusters in Fig. 4b forming key components of two larger clusters; when $\text{threshold}_s$ is large (0.8), the network is sparser, with only the most highly similar nodes of the two clusters forming smaller communities. We thus concluded that the two identified clusters in Fig. 5b were not sensitive to the cosine coefficient threshold, as they were recurrently identified in both smaller and larger cutoff scenarios.

### Continuous structure validation

We validated the consistency of the two clusters' corresponding continuous scores in all IBD cohorts, non-IBD, and healthy control samples, as well as a randomly permuted mock study (as a negative control). The reproducibility of each continuous score within a study was defined as the maximum absolute cosine coefficient between the score's consensus loading (as provided by MMUPHin_Continuous) and the top three principal component loadings discovered independently within that study. Note that the number of top principal components considered here was set to a fixed value (three) instead of based on a percent variance cutoff as in MMUPHin_Continuous. This is because in the two identified clusters in Fig. 5c, the latest included node was PC3. The randomly permuted study consisted of 473 samples (median validation data sets sample size) randomly selected from the entire meta-analysis collection, but each sample's microbial abundance was independently permuted across features. This was to simulate a "negative control" dataset where there should be no continuous population structures.

As we show in Additional file 1: Fig. S12, the dysbiosis score was well validated across studies, except for healthy control samples and the negative control dataset. The Firmicutes-versus-Bacteroidetes tradeoff score, on the other hand, was reasonably well reproduced in all studies and particularly well-established in healthy samples, but, again, was not significantly detected in the negative control dataset.

### *Continuous score assignment*

Assignment of continuous scores was straightforward given the two consensus loading vectors provided by MMUPHin. Within each study, arcsin square root-transformed relative abundances were centered per-feature, the transformed abundance matrix was then multiplied by each consensus loading via dot product to generate per-sample continuous scores. These scores were used for visualization as in Fig. 4d and Additional file 1: Fig. S10, as well as for testing the difference between CD, UC, non-IBD, and healthy control populations as in Additional file 1: Fig. S16. We provide the two consensus loadings in Additional file 9: Table S7; interested researchers can follow these steps to assign the two continuous scores in other datasets.

## Supplementary Information

---

**Additional file 1: Supplemental Figures.**

**Additional file 2: Supplemental Notes.**

**Additional file 3: Supplemental Table 1.** Additional demographic, clinical, and bioinformatics characteristics of included studies.

**Additional file 4: Supplemental Table 2.** Full set of varying simulation parameters evaluated for MMUPHin validation.

**Additional file 5: Supplemental Table 3.** Detailed specification for omnibus and per-feature differential abundance testing models.

**Additional file 6: Supplemental Table 4.** Per-study and meta-analytically aggregated effect sizes of per-feature differential abundance tests performed.

**Additional file 7: Supplemental Table 5.** Per-study and meta-analytically aggregated effect sizes of tests on per-feature abundance associated with disease phenotypic severity.

**Additional file 8: Supplemental Table 6.** Per-study and meta-analytically aggregated effect sizes of tests on the interaction between disease/treatment and sample type/disease on per-feature abundances.

**Additional file 9: Supplemental Table 7.** Consensus loadings for the two identified continuous scores characterizing the IBD gut microbiome population structure.

**Additional file 10: Review history.**

---

**Review history**
The review history is available as Additional file 10.

**Peer review information**

**Authors' contributions**
RX and CH performed conceptualized and determined the study design of this project. SM, RK, and HV acquired the IBD datasets and curated metadata. SM performed the uniform bioinformatics profiling of 16S sequences across studies. SM, DS, and HM developed the methodology for MMUPHin. SM performed the implementation, simulation validation, and IBD application analyses. MS provided interpretations for the meta-analyzed consistent microbial associations. LN provided input on the disease severity / treatment interaction analysis. SM and CH drafted the manuscript. EF, HV, RX, and CH performed substantive text revision. All authors have read and approved the submitted manuscript.

**Funding**
This project is funded by NIH NIDDK R24DK110499 and P30DK043351.

**Availability of data and materials**
IBD datasets used in this study (OTU count profiles, sample metadata) are available at the Human Microbial Bioactives Resource Portal (http://portal.microbiome-bioactives.org). The MMUPHin software [36] is available at Bioconductor. Analysis code is available as a GitHub repository [68]. The version of the software used in the manuscript can be found at Zenodo [69]. All software and code are freely available under the MIT license.

## Declarations

**Ethics approval and consent to participate**
Not applicable.

**Consent for publication**
Not applicable.

**Competing interests**
CH is on the Scientific Advisory Board for Seres Therapeutics and Empress Therapeutics.

**Author details**
[1]Harvard Chan Microbiome in Public Health Center, Harvard T.H. Chan School of Public Health, Boston, MA, USA. [2]Broad Institute of MIT and Harvard, Cambridge, MA, USA. [3]Massachusetts General Hospital, Boston, MA, USA.

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

## 
