## [**Additional file 10: Review history.** · Genome Biology]

Review History

First round of review

Reviewer 1

Are you able to assess all statistics in the manuscript, including the appropriateness of statistical tests used? Yes. My statistical background is limited, and so I would recommend including review from a biostatistician

Comments to author:

The overall goal of this manuscript is to describe a novel computational method for performing meta-analysis of microbiome datasets. Given the continued expansion of such datasets, I believe that this meets a current need in the field. Moreover, I believe that the current tools produced by the Huttenhower group (MetaPhlAn2 and HUMAnN2) are used in the plurality (if not the majority) of such microbiome analyses. Therefore, a tool which can integrate this type of data across cohorts would be a convenient approach for the community of microbiome scientists who are already using this overall computational approach.

Given this background in my understanding of the microbiome field, I was somewhat surprised to see that this particular effort is focused on the analysis of 16S datasets instead of the WGS datasets that previous innovations by the Huttenhower group have focused on. Of course, my reaction is more of a reflection of my own biases than a judgement on the manuscript itself.

Diving into the methods used to process the raw data, I was surprised to see an overall approach to 16S analysis which did not include recent advances in denoising sequencing error, such as has most notably introduced by Ben Callahan and colleagues with *dada2* and subsequently adopted and extended by other groups. Instead, this study takes the approach of OTU clustering at a fixed identity threshold and taxonomic identification by alignment to Greengenes. While I am not an expert on 16S algorithms, this did make me wonder whether the performance of the novel approach being described in this manuscript would be similar with more robust ASV-based approaches.

At the end of the day, the 16S analysis used by the authors generated information in terms of relative abundance of taxonomic groupings. Since that data type is generally analogous to the output of MetaPhlAn2, it may be reasonable to hypothesize that this new approach may be applicable to WGS data as well as 16S.

When reading through the statistical approach used to correct for batch effects and make robust comparisons across studies, I find myself as a computational biologist largely unprepared to critically examine the underlying assumptions which drive the exact choice of methods and approach. While one conclusion to draw from this observation is that it would be helpful to also include a review from a biostatistician, I would also anticipate that many of the intended readers of this manuscript would find themselves in a similar position to myself.

To help convey the value of this statistical approach, I would recommend a visual display of the

novel organisms that were identified as being associated in the IBD meta-analysis. For example, the manuscript could include a figure showing the relative abundance of *Acinetobacter* and *Turicibacter* across the entire datasets, and which compares the normalization of MMUPHin to ComBat and quantile normalization. In other words, when I can't evaluate the statistical approach from first principles, instead I can evaluate how convincing the results are in terms of the novel findings that it provides. I would expect that in this display it would be visually obvious that MMUPHin is appropriately correcting for batch effects while also identifying an association with IBD that would not have been obvious without that normalization approach.

On a similar note, a simple visual display of the "dysbiosis score" as a function of IBD diagnosis status across datasets would be very helpful. I do see that there is a supplemental figure showing the overall "dysbiosis" by IBD status, but it does not convey the batch correction or other features of MMUPHin which are expected to help improve the outcome of the analysis.

My overall feedback on this manuscript is that I did not find myself particularly convinced that this novel methodology would help provide more useful or convincing results from real-world datasets. I would vastly prefer to see a comparison of MMUPHin to existing methods in evaluating the results of the IBD meta-analysis, rather than include so much detail on simulation exercises in the main figures. To be clear, I am not saying that this approach is not valid or useful, but rather that as a reader I don't find myself able to point to any results or figures which demonstrate that superiority clearly.

Summarizing the above narrative in numbered points to help organize feedback:

1. Can you comment on the choice of using OTU-based 16S analysis for this study, rather than ASVs?
2. Would it be possible to include figures in the manuscript which show the relative abundance values of the novel IBD-associated taxa, and demonstrate how their identification would not have been possible with other available methods?
3. Would it be possible to include figures in the manuscript which show that the novel "dysbiosis" score is similarly enabled by the specific advances made in MMUPHin compared to other existing approaches?
4. With the recent publication of SIAMCAT for microbiome meta-analysis, it may be useful to comment (in the text if appropriate) on how MMUPHin compares to the normalization approach which may be used there.
5. When reading over other manuscripts in this area, it appears that an existing approach that should be considered for comparison here is the `removeBatchEffect` function from `limma`.

Reviewer 2

Are you able to assess all statistics in the manuscript, including the appropriateness of statistical tests used? Yes. The statistical approach was fine, it's just that the batch/study correction doesn't seem to have worked and the conclusions are therefore not warranted.

Comments to author:

The authors introduce a collection of methods that allow for meta-analysis of microbiome studies. They note that combining microbiome studies, especially of the gut, is difficult because of strong batch and individual effects. They apply their procedure to a collection of IBD studies and find that they can identify individual microbes associated with different disease subtypes and consistently identify a gradient, and in spite of the large quantity of data, they do not identify microbiome-based subtypes.

My primary concern with this paper is that although the method attenuates the differences between the studies, it does not eliminate them. If my understanding of Figure 3 is correct, a substantial fraction of the study- and batch-related variation remains after the procedure has been applied. I agree with the authors that some of the properties of microbiome data make batch correction difficult, and that attenuating this source of variability is an accomplishment. However, I do not think the conclusions the authors draw from their analysis are warranted. As the authors note in the second paragraph of the Results section (lines 79-82) any analysis before batch correction and meta-analysis "can be misleading due to the confounding of cohort structure between studies." Since differences remain after the correction, it seems to me that this is also true of the meta-analysis presented in the paper.

It is possible that I have misunderstood something, in which case the authors should clarify why attenuating and but not eliminating the batch and study effects suffices.

I have several other more minor comments.

General question: Is it possible to describe the batch effect vs. effect of interest for the real data in terms of the batch strength in the simulations? This would help the reader calibrate how effective the method should be on the real data.

Figure 2: The caption for Figure 2 needs to be more informative. I looked at Figure 2 for the first time after reading the Results section (the first reference is on line 108), and even after reading the caption, I couldn't figure out what R^2 on the y-axis was supposed to be. The reader should not have to go to the supplemental materials to parse a figure.

Supplemental figure 2: I would be interested in standard errors (just a naive version would be fine) on this figure. The boxplot seems like a bit of an odd choice here because the points are not drawn from a population and there is therefore no distribution that we are trying to summarize. What we (or at least I) really want to know from this figure is whether the differences in abundance/prevalence are due to chance or not. The differences are often large enough, and I assume the library sizes are big enough, that the standard errors would be quite small relative to the mean abundances, but I'm not sure if that's also true of the prevalences. That all being said, if the standard errors are all tiny compared to the effect sizes it's better to keep the boxplots.

Line 163: Unclear from the text what about the results support the hypothesis of pro-inflammatory aerotolerant clades forming a positive feedback loop.

Line 417 (first displayed equation in "Simulation validation of MMUPHin"): This formula doesn't make sense. If we exponentiate both sides, we see that Y_{ip} is never equal to zero.

When the Bernoulli takes value 0, Y_{ip} is equal to 1. When the Bernoulli value is non-zero, Y_{ip} can be less than 1 if the normal variable takes a negative value. Is this the intended behavior?

A bunch of fiddly comments on "Unsupervised continuous structure discovery":

Line 385-386 (bullet point 2): Is there a default or guidance on how to choose the customizable similarity threshold?

Line 387: The reference for community detection is just to igraph, and it's not clear from the language here what exactly is happening. Is the reader meant to look at the previous paper for the details?

Line 393-4 (point 4): It's not clear to me that it is always possible to have the signs work out. It will depend on the details of the thresholds for the similarity threshold and the denseness of the connectivity for the modules. Suppose we have the following three PCs:

```
[,1] [,2] [,3]
[1,] -1.38650440 -0.3022241 -1.268658799
[2,] 1.72311730 1.6626374 0.518501652
[3,] -0.64458549 -1.1139292 0.281648595
[4,] -0.42806325 -1.7377230 1.285245698
[5,] -0.06428050 -1.2537129 1.099791236
[6,] 1.09050667 1.5585543 -0.158618842
[7,] 0.61818275 -1.0355818 1.973135085
[8,] 1.49500607 0.9051074 0.990755655
[9,] -0.85899293 0.4665796 -1.610858682
[10,] 1.70777175 0.9993662 0.984633074
[11,] 0.94210852 1.1281060 -0.008446498
[12,] 1.05446578 1.0187603 0.467260791
[13,] -0.05649655 0.3119414 -0.388900395
[14,] 1.32647536 -0.1490256 1.716939688
[15,] -0.41847582 -0.4641675 -0.122296172
[16,] -2.02838181 -0.8189566 -1.433360942
[17,] 0.68963249 0.2148923 0.524455833
[18,] -0.80249485 -1.0057786 -0.069795483
[19,] -0.16633605 -0.5568348 0.425177277
[20,] -0.44020648 -0.5360575 0.194269493
```

The cosine similarity between 1 and 2 and 1 and 3 is about .65, but the cosine similarity between 2 and 3 is -.13 (if I did the computations correctly). This would lead to a network where $1 \leftrightarrow 2$, $1 \leftrightarrow 3$ if the cosine similarity threshold is .6, and I assume this would be a densely connected module if the threshold for the fraction of connections is .6 as well (because the module has $2/3 = .67$ possible connections). However, it is not possible to change all the signs in this example so that all pairs of cosine similarities are positive.

This phenomenon is why I asked about the details of the cluster detection and the threshold for the cosine similarity. If these cutoffs are stringent enough, you don't run into problems like this, but if they're a little looser you can.

Line 395: It is not clear to me that the consensus vectors need to be mutually independent, even approximately. Consider the case where all of the studies have the same set of PCs, but the eigenvalues corresponding to the top PCs are the same and the PCs are not identifiable. Then even with infinite data within each study, the PC network will just be connecting random components from the principal plane and there is no guarantee that the consensus components will be orthogonal.

Line 400 (point 6): I think a better way to do this would be to validate by checking the proportion of variance explained by the identified component in the new dataset instead of correlating the identified component with the top principal components in the new dataset. There are two reasons for this: 1 - If the eigenvalues for the top PCs are similar, there is a lack of identifiability in the actual PCs, and you are stacking the deck against yourself by requiring the PCs to correlate exactly. 2 - Even if you don't have the issue with similar eigenvalues/non-identifiability of the PCs, it seems that it should be "good enough" for the component you identified in the training data to explain a lot of variance in the test set. It seems a bit too strict a criterion for the component to have to correlate with the top PC in the test set. If you're getting that sort of correlation anyway it's probably not worth it to change to a new metric, but I think a less stringent criterion is justified.

Reviewer 3

This ms tried to set up a statistical framework for meta-analysis of datasets from various independent studies. Such attempt is urgently needed for the microbiome field. However, the taxon-based approach employed in this ms represents a huge concern that has hampered the progress of human microbiome field for too long.

The first concern is the ignoring of novel bacteria by the taxon-based analysis which is database-dependent. Sequences representing novel members of the gut microbiome will be left aside from any downstream analysis because they have no nearest neighbor in the database and cannot be given a proper taxonomic position. Without a taxonomic name, such bacteria cannot be lumped together with other bacteria having the same name to reduce the dimensionality of the microbiome datasets. If we only focus on bacteria which have been previously characterized but ignore novel ones, we may not move forward on new findings on gut microbiome in human health and diseases.

The second concern is that after removing unclassified bacteria from the datasets, how much of the left datasets can still reflect the true quantitative relationships of all the known bacteria? If the left dataset cannot reflect the ecological structure of the gut microbiome, any findings from such analysis may be spurious and non-reproducible.

The third concern is related with using taxa as a functional unit for association studies. Bacterial species is technically defined as strains which share higher than 70% homology (DNA-DNA hybridization) or higher than 95% ANI (genome sequence comparison). This means that members in the same bacterial species can have up to 30% difference in their genomic sequences. Because of this, members in the same species can change in opposite directions when disease progresses. This means that some bacteria may increase, others may decrease when the disease gets worse or improved, though all these bacteria may belong to the same species. When you move up to genus level, the situation become even worse as members in the same genus can have only 25% genome homology. If bacteria in the same species/genus or any other taxon level change in opposite directions, any attempts in lumping them together will lead to a new variable which no longer reflect any real ecological relationships among individual bacteria. Unfortunately, this ms started to establish the new statistical framework by looking at phylum level variations. Even species is not a unit with enough resolution for functionally dissecting bacteria, going to phylum level is indeed a huge problem. This is why phylum level microbiome signatures have been the least reproducible. Genus level signatures are also very difficult to be reproducible.

This ms used OTUs as the first level organization for data analysis. The resolution of OTUs is roughly at species level. This is not enough. OTUs do not take advantage of the possible subspecies level resolution that the sequences can offer. The ms should use ASVs as the basic units for analysis.

There's also a widely held misunderstanding on 16S rRNA gene sequencing. Many people believe that you can only go down to genus level when you analyze amplicon sequences. This is only true if you want to do taxon-based analysis. Each ASV represent a unique group of bacteria whose genome homology is somewhere between individual strains and species. You don't need a name to be able to follow the behavior of a bacterium. Each AVS with a unique ID represents a unique group of bacteria. By examining the ecological behavior of ASVs, you will be able to identify unique groups of bacteria which are correlated with host phenotypes. the resolution will be the much higher than any database-dependent and taxon-based analysis.

Reviewer #1

The overall goal of this manuscript is to describe a novel computational method for performing meta-analysis of microbiome datasets. Given the continued expansion of such datasets, I believe that this meets a current need in the field. Moreover, I believe that the current tools produced by the Huttenhower group (MetaPhlan2 and HUMAnN2) are used in the plurality (if not the majority) of such microbiome analyses. Therefore, a tool which can integrate this type of data across cohorts would be a convenient approach for the community of microbiome scientists who are already using this overall computational approach.

We thank the reviewer for their encouraging words and thoughtful suggestions about our work. As the reviewer has kindly summarized their comments into numbered points at the end, we also organized our responses accordingly below.

Given this background in my understanding of the microbiome field, I was somewhat surprised to see that this particular effort is focused on the analysis of 16S datasets instead of the WGS datasets that previous innovations by the Huttenhower group have focused on. Of course, my reaction is more of a reflection of my own biases than a judgement on the manuscript itself.

Diving into the methods used to process the raw data, I was surprised to see an overall approach to 16S analysis which did not include recent advances in denoising sequencing error, such as has been most notably introduced by Ben Callahan and colleagues with dada2 and subsequently adopted and extended by other groups. Instead, this study takes the approach of OTU clustering at a fixed identity threshold and taxonomic identification by alignment to Greengenes. While I am not an expert on 16S algorithms, this did make me wonder whether the performance of the novel approach being described in this manuscript would be similar with more robust ASV-based approaches.

At the end of the day, the 16S analysis used by the authors generated information in terms of relative abundance of taxonomic groupings. Since that data type is generally analogous to the output of MetaPhlan2, it may be reasonable to hypothesize that this new approach may be applicable to WGS data as well as 16S.

When reading through the statistical approach used to correct for batch effects and make robust comparisons across studies, I find myself as a computational biologist largely unprepared to critically examine the underlying assumptions which drive the exact choice of methods and approach. While one conclusion to draw from this observation is that it would be helpful to also include a review from a biostatistician, I would also anticipate that many of the intended readers of this manuscript would find themselves in a similar position to myself.

To help convey the value of this statistical approach, I would recommend a visual display of the novel organisms that were identified as being associated in the IBD meta-analysis. For example,

the manuscript could include a figure showing the relative abundance of Acinetobacter and Turicibacter across the entire datasets, and which compares the normalization of MMUPHin to ComBat and quantile normalization. In other words, when I can't evaluate the statistical approach from first principles, instead I can evaluate how convincing the results are in terms of the novel findings that it provides. I would expect that in this display it would be visually obvious that MMUPHin is appropriately correcting for batch effects while also identifying an association with IBD that would not have been obvious without that normalization approach.

On a similar note, a simple visual display of the "dysbiosis score" as a function of IBD diagnosis status across datasets would be very helpful. I do see that there is a supplemental figure showing the overall "dysbiosis" by IBD status, but it does not convey the batch correction or other features of MMUPHin which are expected to help improve the outcome of the analysis.

My overall feedback on this manuscript is that I did not find myself particularly convinced that this novel methodology would help provide more useful or convincing results from real-world datasets. I would vastly prefer to see a comparison of MMUPHin to existing methods in evaluating the results of the IBD meta-analysis, rather than include so much detail on simulation exercises in the main figures. To be clear, I am not saying that this approach is not valid or useful, but rather that as a reader I don't find myself able to point to any results or figures which demonstrate that superiority clearly.

Summarizing the above narrative in numbered points to help organize feedback:

1. Can you comment on the choice of using OTU-based 16S analysis for this study, rather than ASVs?

We fully appreciate the reviewer's comment, and agree that ASVs are the current state-of-art approach for 16S analysis due to their generally improved resolution and ease of use. As is too often the case, the use of OTUs in the current analysis was essentially a historical artifact, as the project began long enough ago that ASVs were not yet well-established. To update this and address the reviewer's concerns, we have performed additional analyses to show that:

- a) Our method can be applied to ASV profiles to reduce batch effects, similar to OTU profiles.
- b) Empirically, OTU- and ASV-oriented pipelines yielded similar taxonomic abundances in our studies, and thus the choice of sequence units had minor impacts in terms of the meta-analysis results.

We first showed that MMUPHin's batch correction method effectively reduces technical effects in real-world microbial ASV profiles (new **Supplemental Fig. 18a**). We applied the dada2 method[1] to generate ASV abundance tables on two of the studies included in our meta-analysis: BIDMC-FMT as a case with small sample size (n=16) and two batches, and RISK with large sample size (n=882) and multiple batches (15 total). We applied MMUPHin to perform batch normalization on the two studies separately, and evaluate the total ASV abundance variability in either dataset attributable to batch effects before and

after normalization. As presented in the new **Supplemental Fig. 18**, MMUPHIn effectively reduced batch variability in the resulting ASV-based microbial abundance profiles.

New Supplemental Figure 18: MMUPHIn meta-analysis method and results apply comparably to OTU and ASV abundance profiles. a) MMUPHIn batch correction successfully reduces batch difference in ASV abundance profiles, in both small and large sample size studies with different numbers of technical batches. Batch effects in either study are quantified through PERMANOVA R² as in Figure 3a. **b)** When aggregated at the genus level, the choice of OTU- versus ASV-based bioinformatics pipelines has limited impact on the generated abundance profiles, given properly configured and quality-controlled OTU formation parameters (as previously described[2]). Pipeline effect is again quantified through PERMANOVA R² statistics.

We additionally evaluated that, when aggregated at the genus level (which was used as the taxonomy level for all analyses in our paper to ensure compatibility between prior studies), OTU-based and ASV-based pipelines generated very similar abundance profiles (new **Supplemental Fig. 18b**). Sequence units, either clustered OTUs or individual ASVs, were annotated using Greengenes version 13.8, and then aggregated at the genus level. We then visualized the difference in the two bioinformatics pipelines in ordination space, and quantitatively evaluated this effect, i.e., the variability in the combined abundance profiles attributable to pipeline difference, with PERMANOVA R² statistics. While there is indeed an effect due to the difference in bioinformatics pipelines, it is limited when taxa are aggregated at the genus level.

The above new evaluations, along with the new **Supplemental Fig. 18**, are consolidated into the following paragraph in **Methods**:

“We additionally evaluated the impact of OTU- versus ASV-based bioinformatics pipelines on our method and meta-analysis results (**Supplemental Fig. 18**). Specifically, we processed two studies representing extremes of size, BIDMC-FMT (n=16, two technical batches) and RISK (n=882, fifteen batches) with the dada2 method[1], and performed batch-correction and evaluation on the generated ASV profiles. MMUPHIn was still capable of reducing batch effects in either study’s ASV-based abundance profiles, showcasing that our method is applicable to such new bioinformatics protocols (**Supplemental Fig. 18a**). Additionally, when aggregated at the genus level, OTU- and ASV-based abundance profiles had limited differences, suggesting that the choice of sequence variant units has limited impact on our meta-analysis results, as previously indicated[2] (**Supplemental Fig. 18b**).”

2. Would it be possible to include figures in the manuscript which show the relative abundance values of the novel IBD-associated taxa, and demonstrate how their identification would not have been possible with other available methods?

The reviewer raised several good points in this thread, which we address in three parts.

First, to the author's point regarding if *Turicibacter* and *Acinetobacter* were uniquely identified by MMUPHin: these initially caught our eye due to their underrepresentation in shotgun metagenomic profiles relative to 16S amplicons, caused by them being 1) poorly captured by isolate reference genome databases (and thus many profiling tools, prior to recent metagenome assembly efforts) and 2) quite low abundance in typical fecal communities (and thus below the sensitivity of many metagenomic, as compared to 16S, profiles). This raised their novelty in a 16S-based meta-analysis in the first place, and they were not individually highlighted by any of the previous, individual studies that we integrated here.

Given this, to differentiate their novelty from increased sample size vs. improved batch correction, we ran our meta-analysis using only ComBat-corrected data, and found that *Turicibacter* was also significant ($p = 0.001$) whereas *Acinetobacter* was marginally non-significant ($p = 0.0635$). Thus the discovery of these new associations is arguably due mainly to the improved sample size of combining multiple meta-analysis studies, with accompanying data-appropriate bioinformatics and statistical pipelines, plus a smaller additional improvement specifically due to MMUPHin's more tailored batch correction method.

We further expanded this new analysis to more systematically compare discoveries possible within this dataset using MMUPHin vs. simpler (or no) batch correction methods. When doing so in real data, of course, the ground truth of which taxa are truly associated with the disease is unknown, thus both agreement with alternative methods and method-unique discoveries are often used to measure performance[3]. To this end, we compared the reported significant genera from ComBat, quantile normalization, and MMUPHin in IBD versus control testing (new **Supplemental Fig. 11**). For each method's normalized microbial profiles, we pooled samples across studies to perform differential abundance testing on IBD status. For ComBat and MMUPHin, two-sample t-tests were performed on arcsine square-root transformed relative abundances, to provide as simple a baseline as possible and isolate the effects of batch correction. For quantile normalization, the normalized profiles are percentiles and not appropriate for t-testing, and we adopted Mann-Whitney nonparametric testing as the closest simple analog[4].

Across the three approaches, MMUPHin identified the most significantly different genera ($p < 0.05$), and was in most agreement with either alternative (new **Supplemental Fig. 11**).

New Supplemental Figure 11: MMUPHin batch correction outperforms quantile normalization and ComBat in real-world IBD association analysis. Venn diagram indicates the number of features identified individually or jointly after batch correction by quantile normalization, ComBat, or MMUPHin when comparing IBD versus control microbial profiles. For quantile normalization, univariate nonparametric Mann-Whitney tests were performed for the normalized percentiles pooled across studies, as recommended in [4]. For ComBat and MMUPHin corrected microbial relative abundances, univariate two-sample t-tests were performed for consistency with the quantile normalization analysis. MMUPHin correction identified the greatest number of significant genera ($p < 0.05$) out of the three approaches. Additionally, it showed the largest agreement with both of its alternatives.

We would note that, in the absence of additional experimental work, it is difficult to more quantitatively establish that MMUPHin results are “better” at identifying true IBD associations - hence our focus on extent and concordance of its results. Indeed, to make this comparison more accurate from a technical perspective, the tests performed do not account for factors that would likely matter biologically, e.g. additional clinical and demographic covariates. The main reason is practical: the quantile normalization publication noted their normalized profiles are only appropriate for pooled univariate case-versus-control testing. Thus to ensure comparability between the three methods, we performed consistent univariate testing for each. The results should thus only be interpreted as performance evaluations of the methods. For biological interpretations, readers should still refer to our originally reported multivariate analysis results (**Figure 3B**). To show and explain these analyses, the following text, along with the new supplemental figure, has been added to our **Results** section:

“Lastly, we also conducted a more direct comparison of IBD microbiome associations found after applying each of the three batch correction methods (quantile normalization, ComBat, and MMUPHin) to our meta-analysis dataset. This employed a simpler post-correction testing strategy, as previously recommended[4], thus making the results more directly comparable but likely less biologically relevant than those discussed previously (**Fig. 3**). MMUPHin-corrected abundance profiles still identified more IBD-associated genera compared to ComBat and quantile normalization, while also showing the best

agreement with both other methods (**Supplemental Fig. 11**). This provides empirical evidence for MMUPHin's effectiveness in real-world settings, in addition to its accuracy as quantified by our simulation studies."

Last, we are hesitant to agree with the reviewer's comment that simulation results are less important to validate MMUPHin's performance. In general, we aim to always pair some type of synthetic evaluation with one or more real-world validations during any type of methods development, since the two approaches provide such nicely-complementary information. While real-world results typically provide more interpretable empirical evidence, the ground truth of which taxa are biologically differential is unknown. Thus, even if a certain method identified more significant associations, it is impossible to tell if this indicates better power or inflated false positive rates, without experimental validations that are often prohibitive. Our simulation-based analyses, instead, a) generated microbial abundance profiles specifically tailored to closely resemble those of real data[5], b) adopted a "neutral" model that does not necessarily favor MMUPHin, ComBat, or quantile normalization (**Methods**), and c) importantly, yielded both differentially abundant and null microbes that are known *a priori*, which then form the "gold standard" for computing common metrics such as false positive rate and power. Simulation-based analyses are widely adopted in practice to validate methods, often preceding real-world results[6, 7]. We hope that, in combination with our existing and newly provided real-world analysis results, they will be convincing to the reviewer and general readers on MMUPHin's performance.

To more clearly convey our goals with these joint evaluations, the following sentence has been updated to the simulation section in **Results**:

"These simulations were designed to be complementary to our application to and assessment of the IBD microbiome as described below, since they allow analysis of a controlled ground truth of outcome-associated and null microbial elements that is lacking in uncontrolled population settings. As detailed in **Methods**, our simulation approach a) generates realistic microbial profiles, so that the evaluation findings are generalizable to the appropriate target populations, and b) is neutral to the evaluated methods (ComBat, quantile normalization, MMUPHin, etc.)"

Would it be possible to include figures in the manuscript which show that the novel "dysbiosis" score is similarly enabled by the specific advances made in MMUPHin compared to other existing approaches?

If we understand this comment correctly, we may have presented the "dysbiosis axis" finding poorly, since it is not a predictive model or differential abundance finding that can be meaningfully compared with different methods. Specifically, as per the bottom of **Fig. 1A**, MMUPHin provides two analysis methods after batch correction: differential abundance testing and population structure discovery. While for association testing it was clearly meaningful to compare the profiles as corrected by different approaches, our dysbiosis scores were identified by MMUPHin's own continuous structure discovery method, which, importantly, corrects for per-study batch effects in overall population structures with its PC network approach. We were thus unclear on how to compare against an alternatively identified

“dysbiosis score”, as to our knowledge MMUPHin is the only existing option for meta-analyzed continuous population structure discovery across microbiomes. Conversely, comparison of the final joint dysbiosis axis to previous individual studies (e.g. [8]) is already implicitly carried out as part of the method, and MMUPHin’s result thus subsumes the individual studies’ results by definition. We are of course happy to further expand or clarify this, if we were off-target in our interpretation here.

With the recent publication of SIAMCAT for microbiome meta-analysis, it may be useful to comment (in the text if appropriate) on how MMUPHin compares to the normalization approach which may be used there.

We thank the reviewer for calling out this method, which was published during the submission of MMUPHin (which, again, was delayed for logistical reasons). We have now added a direct comparison of MMUPHin’s differential abundance findings on our joint IBD dataset with the equivalent differences found by SIAMCAT, described below.

We first note, however, that this comparison is not 100% appropriate, since SIAMCAT is focused specifically on predictive models in the context of multiple studies, as distinct from the batch correction and joint linear modeling carried out here. While the term “meta-analysis” can be used in both cases, 1) SIAMCAT does not carry out any type of within-study batch normalization or between-study harmonization, and 2) each predictive model applies only to individual datasets (and are aggregated later on). While it would in theory be possible to apply SIAMCAT (or any machine learner) to a joint dataset post-batch-correction, their recommended protocol specifically states not to.

In particular, as described by the accompanying vignette for the software (https://bioconductor.org/packages/release/bioc/vignettes/SIAMCAT/inst/doc/SIAMCAT_meta.html), a) their normalization approaches carry out only per-study transformation, such as log transformation and distribution standardization, to prepare for machine learning analysis, and b) their differential abundance testing functionality is limited, in that only per-study analysis is performed with no aggregation to generate meta-analysis p-values equivalent with those reported by MMUPHin.

Given these differences, we thus focused on comparing highly-weighted features in SIAMCAT’s machine learning module with the taxa reported by MMUPHin to be significantly differential. We trained SIAMCAT models on all individual studies where both IBD and control samples were available. Each feature’s median relative importances were then averaged across studies to compare against their significance levels ($q < 0.05$) by MMUPHin. MMUPHin significant features had higher absolute average variable importance as reported by SIAMCAT (Mann-Whitney $p = 1.2e-5$, new **Supplemental Fig. 10**). We also manually verified that the remaining differences between MMUPHin and SIAMCAT are due to the different focuses of the two methods (i.e. joint prediction of SIAMCAT versus per-feature testing of MMUPHin), and are happy to provide further details if helpful.

New Supplemental Fig. 10: MMUPHin meta-analysis findings agree with SIAMCAT machine learning feature importances. We trained SIAMCAT machine learning models predicting IBD status in each individual study where both IBD and control samples were available. From the per-study trained models, each genus's median relative importances were averaged. These were then compared against genera significance levels ($q < 0.05$) as identified by MMUPHin. MMUPHin significant features had higher absolute average variable importance as reported by SIAMCAT ($p=1.2e-5$, one-sided Mann-Whitney U test).

The following additional comment was added to the **Results** section:

As expected, our meta-analysis confirms many of the taxa associated with IBD reported by previous individual studies (**Fig. 3b**, detailed in **Supplemental Notes**); *“they also agreed with important features as identified through other types of predictive, rather than hypothesis testing, machine learning models (Supplemental Fig. 10[9]).”*

When reading over other manuscripts in this area, it appears that an existing approach that should be considered for comparison here is the `removeBatchEffect` function from `limma`.

This is an interesting suggestion, but to our understanding, `removeBatchEffect` is already a simpler version of `ComBat`, as it adopts the same linear modelling paradigm but does not perform the empirical Bayes shrinkage on per-batch parameters as in `ComBat` and `MMUPHin`. This was supported by its behavior during an additional evaluation of its performance on our simulated data, which showed that while `removeBatchEffect` also reduces batch effects, it did not retain biological signals as did `ComBat` and `MMUPHin` (**Response Fig. 1**). Given these two considerations - its relatively poor performance, and its conceptual similarity to `ComBat` - we currently show these results only in our response, although we are happy to include them in the manuscript's supplement if it would be helpful.

Response Figure 1: limma's removeBatchEffect correction is outperformed by ComBat and MMUPHin. We evaluate the performance of limma's removeBatchEffect function in the same simulation analysis as for ComBat and MMUPHin (Fig. 2A). limma was indeed able to reduce batch effects, although to a substantially lesser degree than ComBat and MMUPHin. Critically, however, it was also unable to retain the effect of positive control variables while doing so.

Reviewer #2

The authors introduce a collection of methods that allow for meta-analysis of microbiome studies. They note that combining microbiome studies, especially of the gut, is difficult because of strong batch and individual effects. They apply their procedure to a collection of IBD studies and find that they can identify individual microbes associated with different disease subtypes and consistently identify a gradient, and in spite of the large quantity of data, they do not identify microbiome-based subtypes.

My primary concern with this paper is that although the method attenuates the differences between the studies, it does not not eliminate them. If my understanding of Figure 3 is correct, a substantial fraction of the study- and batch-related variation remains after the procedure has been applied. I agree with the authors that some of the properties of microbiome data make batch correction difficult, and that attenuating this source of variability is an accomplishment. However, I do not think the conclusions the authors draw from their analysis are warranted. As the authors note in the second paragraph of the Results section (lines 79-82) any analysis before batch correction and meta-analysis "can be misleading due to the confounding of cohort structure between studies." Since differences remain after the correction, it seems to me that this is also true of the meta-analysis presented in the paper.

It is possible that I have misunderstood something, in which case the authors should clarify why attenuating and but not eliminating the batch and study effects suffices.

Many thanks for the reviewer's time and input on the manuscript, and we agree that MMUPHin, in most cases, reduces but does not completely eliminate batch effects within or between microbiome profiles. However, this is as expected - much as we might prefer otherwise! - and definitely does not impede the utility or applicability of the method. Notably, it occurs for fairly straightforward reasons (briefly, the loss of information due to non-detections in the underlying measurement platforms, and a preference for variance rather than bias during shrinkage), and is present (albeit to a much lesser degree) in other types of batch correction for molecular data, such as transcriptional profiling (and thus ComBat itself).

First, and perhaps most importantly, residual batch effects do not prevent analysis after their reduction; indeed, reduction of batch effects improves meta-analysis results to the extent that it decreases their effect size relative to that of biological outcomes and covariates of interest (as long as additional bias or error is not introduced in the process). By reducing the effect size of batch differences, even if they cannot be eliminated, MMUPHin (and other methods) enable more (and more accurate) analyses than would have been possible otherwise.

Further, MMUPHin's downstream analysis steps are not directly dependent on its batch normalization process. For per-feature differential abundance testing, we adopt linear mixed effects models, which can explicitly account for residual per-study effect differences. For unsupervised population structure discovery, we perform PC network clustering approach (discussed in more details below in response to

the reviewer's additional comments) to prioritize consistent biological signals over batch effects. The validity of these methods - even in the presence of residual batch effects - is quantified through our simulation studies. For example, evaluation of differential abundance testing establishes that MMUPHin successfully controls false positive rates with good power (**Figure 2c-d, Supplemental Fig. 7**). In real data analysis, the fact that the majority of MMUPHin-identified disease and treatment associations confirm previous findings also provide confidence for these results.

Second, this behavior is common to most batch correction methods applied to real-world molecular data; very similar results are clearly visible in, for example, Figures 3 and 4 of the recent ComBat-seq publication[10] or panel C from most figures in a recent evaluation of scRNA-seq batch correction approaches[11]. The tradeoff is between reducing batch effects as much as is practical, while not also eliminating "real" biological information at the same time. Each approach tends to make its own assumptions and tradeoffs in this regard, with MMUPHin's detailed below.

Third, we intentionally designed MMUPHin conservatively, i.e. to incompletely remove batch effects when this is the most analytically desirable outcome. There are two motivations for this:

- A) For microbiome data specifically, MMUPHin only corrects non-zero abundances, because it is unclear to us how to (or even if one should) correct batch differences with respect to (non-)detection of microbial features (rather than their abundance once detected). For a microbe that has greater prevalence in a particular batch than others, it is difficult for us to justify "correcting" its present, measured values to absences (zeros), given that these are unlikely to be technical errors if the bioinformatics were performed with care, and the non-zero sequencing counts are often substantial. The opposite scenario, imputing abundance for batch-induced absence, is practically more reasonable, but has obvious technical difficulties (data imputation - creating measurements out of nothing - is at best a different problem).

Specifically, for both ComBat and our method, correcting batch effects means removing the batch-related signals (location and scale terms γ_{ip} and δ_{ip} in our model specification), while maintaining both the biological signal β_p and the per batch-sample-feature "noise" ϵ_{ijp} . The latter is important because it might contain information from confounders unmeasured in the study design or not included in the batch correction model. Problems arise when one tries to "maintain" ϵ_{ijp} for absent microbes/samples. While for continuous abundances, we estimate ϵ_{ijp} by subtracting the biological and batch signals, this term is difficult to estimate for zero-values, which can be viewed as censored values (and thus essentially a mathematical integral) and lose relevant information in the continuous ϵ_{ijp} . Ideally, correcting presence/absence batch differences could also be addressed for microbiome data, which have significant sparsity, but due to the aforementioned challenges, we adopted a more conservative philosophy with MMUPHin, which only corrects non-zero values and thus cannot fully eliminate batch effects.

- B) From a high-dimensional modelling perspective, MMUPHin inherits ComBat's philosophy, which intentionally shrinks estimates for batch effects towards null with the empirical Bayes design.

One can see this reasoning via the variance-bias trade-off argument. For transcriptome/microbiome data, the number of features often surpass sample size, which means the per-batch/microbe batch effect parameters are high-dimensional. The empirical Bayes shrinkage thus acts as a regularizer, preventing the model from “over-fitting” batch differences when sample size is small and/or there are many batches/studies.

To summarize this combination of information from the previous batch correction literature, our own methodological assumptions, and their ramifications (or, more often, lack thereof) on downstream analyses, we have added the following text to our **Methods** section. We additionally made MMUPHin’s design to only correct non-zero abundance batch effects more explicit in the same **Methods** section:

“With this model specification, we expect MMUPHin to often reduce, rather than fully correct batch differences. This is due to two considerations. First, MMUPHin focuses on correcting non-zero abundance batch effects, and does not change features’ presence/absence across batches. “Correcting” a feature’s batch-specific presence to absence is inappropriate, as substantial non-zero read counts indicate biological presence rather than technical artifacts. Imputing non-zero abundance for batch-specific absence is technically challenging in our linear modelling framework, as the per-sample/feature noise ϵ_{ijp} cannot be reliably inferred for inflated zero values. Second, the empirical Bayes batch effect estimates $\widehat{\gamma}_{ip}^*$ and $\widehat{\sigma}_{ip}^*$ are shrunken from their frequentist counterparts, which provides regularization for high-dimensional parameters as in ComBat, and avoids “overfitting” to batch differences in small sample sizes. MMUPHin’s design is thus intentionally conservative, by correcting batch differences that can be confidently inferred, and maintaining those that are not (which thus also avoids eliminating non-batch, biological signal).”

To make it clear that MMUPHin controls for batch differences not only through its correction method, but also in other analysis components, the following text has also been added to our **Discussion** section:

Reducing inter-study variation in microbial community profiles is challenging relative to other 'omics data types due to 1) the extreme heterogeneity of microbes within most communities (exacerbating both technical and biological differences), and 2) feature zero-inflation arising from both biological and technical reasons. *“MMUPHin alleviates this problem by taking care to incorporate batch/study effects in each of its components, not just during batch effect normalization: mixed effects modelling was adopted for differential abundance meta-analysis to allow for residual per-study effects; PC network clustering prioritizes consistent biological signals over batch effects. Thus despite these challenges, ”* MMUPHin was able to meta-analyze amplicon profiles in this study both to associate microbial shifts with disease outcome, to associate them with treatment-specific differences, and to identify a single pattern of typical microbial variation within IBD.

I have several other more minor comments.

General question: Is it possible to describe the batch effect vs. effect of interest for the real data in terms of the batch strength in the simulations? This would help the reader calibrate how effective the method should be on the real data.

Among the simulation results, the scenario best matched to the characteristics of our collection of real-world data has eight batches and four thousand samples in total (versus ten real studies and 4,789 samples post filtering), 200 microbial features (249 genera), with 10% spiked features at batch effect size 10, which yields ~10% PERMANOVA R^2 corresponding to to batch effect and 3% R^2 corresponding to binary exposure effect (10.98% for studies and 3.48% for sample type observed in real data). This set of results is presented in the rightmost column of second rows of **Supplemental Fig. 6's** panels, and we have updated its caption to include this information:

Supplemental Fig. 6: Full set of performance evaluation and comparison of MMUPHin's batch adjustment method. a-d: Panels are organized by variables (batch, binary positive control, continuous positive control, and negative control) evaluated by the PERMANOVA R^2 . For **a-d**, the panel at second row, rightmost column corresponds most closely with our collection of real-world studies for meta-analysis, in terms of data characteristics : eight batches and four thousand samples in total versus ten real studies and 4,789 samples post filtering; 200 microbial features versus 249 real genera; 10% spiked features at batch effect size 10 that yielded ~10% PERMANOVA R^2 for batch effect and 3% R^2 for binary exposure, versus 10.98% for studies and 3.48% for sample type observed in real data.

It is difficult to provide readers with such a calibration in the completely general case, since the way in which effect sizes are specified for our simulation (in which underlying, true abundances and associations are known) is different than the way in which they are assayed in real data. Thus a mapping can be calculated empirically for any given setting, but not in an easily automatable manner. We are, however, working to add this feature to the underlying simulation framework (SparseDOSSA[12]) in future versions. In the context of MMUPHin, we've explained this with the following text added to the manuscript's **Methods** section:

"To relate these real data results to our simulation evaluation, the real-world data characteristics best correspond with the simulated scenario with eight batches and four thousand samples in total (versus ten real studies and 4,789 samples post filtering), 200 microbial features (249 real genera), and 10% spiked features at batch effect size 10, which yielded ~10% PERMANOVA R^2 for batch effect and 3% R^2 for binary exposure (10.98% for studies and 3.48% for sample type observed in real data) (**Supplemental Fig. 6** panels, second row, last column.)."

Figure 2: The caption for Figure 2 needs to be more informative. I looked at Figure 2 for the first time after reading the Results section (the first reference is on line 108), and even after reading the caption, I couldn't figure out what R^2 on the y-axis was supposed to be. The reader should not have to go to the supplemental materials to parse a figure.

We apologize for the insufficient details in **Figure 2** caption, which is relatively standard in the field (e.g. Fig. 4 in 33015620, Fig. 2 in 27126039). This has been expanded, for both the R^2 explanation and other panels, into the following::

“Figure 2: Effectiveness of batch correction, association meta-analysis, and unsupervised population structure discovery methods. All evaluations use simulated microbial community profiles as detailed in Methods. Left panels summarize representative subsets of results (full set of simulation cases presented in **Supplemental Table 2** and results in **Supplemental Fig. 6-9**), and right panels show examples of batch-influenced data pre- and post-correction. **a, b)** MMUPHin is effective for covariate-adjusted batch effect reduction while maintaining the effect of positive control variables. *For panel a), PERMANOVA R^2 statistics summarize the effect of batch and positive/negative control variables on the overall microbial composition, before and after batch correction.* Results shown correspond to the subset of details in **Supplemental Fig. 6** with number of samples per batch = 500, number of batches = 4, and number of features = 1000 with 5% spiked with associations. **c, d)** Batch correction and meta-analysis reduces false positives when an exposure is spuriously associated with microbiome features due to an imbalanced distribution between batches. Corresponds to **Supplemental Fig. 7** with number of samples per batch = 500, number of features = 1000 with 5% spiked associations, and case proportion difference between batches = 0.8. Evaluations of BDMMA generates low FPRs due to the zero-inflated nature of simulated microbial abundances, and are included only in **Supplemental Fig. 7**. **e, f)** Batch correction improves correct identification of the true underlying number of clusters during discrete population structure discovery. *Success rate is measured as the percentage of selecting the true number of clusters (before and after correction) across simulation iterations.* Corresponds to **Supplemental Fig. 8** with number of batches = 4. **g, h)** Continuous structure discovery accurately recovers microbiome compositional gradients in a simulated population. *We compare identified continuous structure loading with true scores with Pearson correlations.* Corresponds to **Supplemental Fig. 9** with number of batches = 6.”

Supplemental figure 2: I would be interested in standard errors (just a naive version would be fine) on this figure. The boxplot seems like a bit of an odd choice here because the points are not drawn from a population and there is therefore no distribution that we are trying to summarize. What we (or at least I) really want to know from this figure is whether the differences in abundance/prevalence are due to chance or not. The differences are often large enough, and I assume the library sizes are big enough, that the standard errors would be quite small relative to the mean abundances, but I'm not sure if that's also true of the prevalences. That all being said, if the standard errors are all tiny compared to the effect sizes it's better to keep the boxplots.

We agree with the reviewer’s suggestion. In the new **Supplemental Fig. 4** we examined the top five features for each panel (i.e., abundance of highly-prevalent features, prevalence of medium-prevalent features). The study differences in these characteristics far exceed those expected from the per-study standard errors. We thus conclude that the batch differences in mean abundances and prevalences indeed greatly exceed chance, as expected. The new Supplemental Figure is now included in our revised manuscript:

New Supplemental Fig. 4: Observed differences in per-study mean abundance and prevalence exceed those expected by chance. We examined the top features in **Supplemental Figure 2** for the standard error in their per-study mean abundance and prevalence to ensure that the observed study differences are not due to chance. **a)** The spread across studies of per-study mean abundances far exceeds those expected from their standard errors. As in **Supplemental Figure 2a**, each point represents a study-specific mean abundance of one feature, with bars indicating its standard error. **b)** The spread across studies of per-study prevalence also exceeds that expected from their standard errors. Each point represents a study-specific prevalence of one feature, with bars indicating standard errors.

Line 163: Unclear from the text what about the results support the hypothesis of pro-inflammatory aerotolerant clades forming a positive feedback loop.

This refers to a current topic in the IBD microbiome literature, initially introduced in [13] (and similarly in [14] for T2D), arguing that local and systemic inflammation both affect and are affected by the gut microbiome in specific functional ways. In response to a host, microbial, or environmental trigger, basal inflammation and immune activation increases; this selects for microbes that are immune-tolerant via specific pathways (aerotolerance, resistance to increased redox stress, metabolism of host-derived glycans and mucin, epithelial localization and binding); these microbes perturb the immune system in such a way as to sustain or increase inflammatory activity; and the cycle reinforces itself. Several microbes enriched in our analysis are common participants in this process, e.g. the Enterobacteriaceae (facultative anaerobes), enterococci, and typically oral *Fusobacterium*, *Dialister*, and *Veillonella*. Conversely, strict anaerobes typical of broadly defined “health” in the gut are depleted (many Lachnospiraceae, Ruminococcaceae, and other Firmicutes clades).

These results are generally well-established in IBD, and thus we do not focus on them extensively, but we have expanded the Results text to better explain this:

“These findings agree with the emerging hypotheses of pro-inflammatory aerotolerant clades (e.g. *Escherichia* and other Enterobacteriaceae) forming a positive feedback loop in the gut during inflammation, often of oral origin[15] (e.g. *Fusobacterium*, *Dialister*, *Veillonella*), and depleting the gut’s typical fastidious anaerobe population as a result (primarily Ruminococcaceae, Lachnospiraceae, and other Clostridia and Firmicutes clades)[8].”

Line 417 (first displayed equation in "Simulation validation of MMUPHin"): This formula doesn't make sense. If we exponentiate both sides, we see that Y_{ip} is never equal to zero. When the Bernoulli takes value 0, Y_{ip} is equal to 1. When the Bernoulli value is non-zero, Y_{ip} can be less than 1 if the normal variable takes a negative value. Is this the intended behavior?

We apologize for the mistake here. Y_{ip} is a mixture of Bernoulli zeros and a log normal distribution. That is:

$$Y_{ip} \sim \text{LogN}(\mu_p, \sigma_p^2) \times \text{Bernoulli}(\pi_p)$$

In practice, the continuous Y_{ip} are rounded to nearest integer values to generate count observations. The equation typo has been updated in the manuscript.

A bunch of fiddly comments on "Unsupervised continuous structure discovery":

The reviewer raised a series of good points here, mostly related to the concern that our current PC network clustering does not necessarily yield consistent clusters, i.e., all loadings within the cluster have positive cosine coefficients. We make our overall response first in three points, and then provide per-comment responses to the specific items below. First, we discuss the theoretical lower bounds of cosine cutoffs that guarantee cluster consistency, and contrast against those empirically observed in this work, which were much more lenient. Second, we provide more details on our adopted cluster detection algorithm, which inherently alleviates the problem by balancing between larger clusters and smaller ones that are better connected. Last, in cases where this does happen, our implementation provides an explicit warning that suggests the user adopt more stringent criteria.

We first note that this component of MMUPHin is based on previously published results using the same technique for transcriptomics[16]. During earlier development of the method, we observed that, empirically, clustered PC loadings tended to distribute in the same direction, even when the cosine thresholds were set much lower than the theoretical bounds required to guarantee consistency (new **Supplemental Fig. 17a**). For a cluster of connected n PC loadings, the lower bound for cosine coefficients that ensures all loadings are not negatively “angled” is $\cos \frac{\pi}{2(n-1)}$. The boundary case happens exactly when all loadings are located on the same two-dimensional plane, and equally spaced with an angle of $\frac{\pi}{2(n-1)}$ in between.

In practice though, we observe that real-world data (both transcriptomic and microbial) are much better behaved than this, essentially never approaching this pathological scenario, and thus warrant more

lenient thresholds. When the threshold was set to 0.6 in our evaluation, the lowest cosine value between PC loadings within a cluster (post direction correction) were all positive, and much higher than the theoretical “worst case scenario”, which is $\cos(\cos^{-1} 0.6 * (n - 1)) \vee -1$ for a cluster of size n by the above reasoning (new **Supplemental Fig. 17a**). While this is an empirical evaluation, it does provide evidence that real-world data with true, recurrent biological signals display better behavior than the theoretical worst case, and thus perform much better using more lenient thresholds.

With any specified cosine threshold, the clustering algorithm adopted by MMUPHIn prevents clusters that are long “chains” (new **Supplemental Fig. 17bc**), and thus reduces inconsistent clustering. For clustering metric, we use igraph to cluster vertices to maximize the modularity score[17], which is defined as:

$$\frac{\#Within - Cluster Edges}{\#Total Edges} - \frac{\sum_{cluster} (total\ degree\ of\ vertices\ in\ the\ cluster)^2}{4(\#Total\ Edges)^2}$$

The first term is maximized when the entire graph is one cluster, whereas the second term is minimized when each vertex forms its own cluster. Thus maximizing this modularity score (difference between the two terms) strikes a balance between identifying large clusters, versus smaller ones that are more densely connected intracluster. As examples of this criteria, a chain of three nodes will be identified as one cluster, but a chain of four will be broken up into two clusters of size two each (new **Supplemental Fig. 17c**). Long chains are more likely to be inconsistent in terms of cosine coefficients, such as in the theoretical worst case discussed above. Our adopted clustering algorithm precludes such results and thus will generate more consistent clusters.

Still, we note that neither of our arguments above guarantees that the identified clusters will always be consistent, as the reviewer correctly noted in their example below. In such cases, we implement error messages in our software to suggest the user to adopt more stringent thresholds.

The above discussions are now included in **Supplemental Materials** along with the new Supplemental Figure. We also expanded the **Methods** section correspondingly. Steps 3 and 4 of our algorithm now include additional details:

3. “In the resulting network, we perform cluster detection based on modularity score[17, 18] to identify densely connected modules of PCs. Each module by definition consists of PCs from different datasets that are similar to each other - whether or not they occur in the same order or with similar percent variance explained - and which thus represent strong feature covariation signals that are recurrent in studies.

- a. *Clustering by modularity score avoids large clusters with few intracluster edges, and prioritizes smaller clusters that are more densely connected (**Supplemental Fig. 17, Supplemental Materials**). This is relevant for MMUPH in because the more densely connected a cluster is, the better consistency the PCs in the cluster have, which provides evidence for recurring biological signals across the spanned datasets.*
4. For a module k containing PC set M_k , its consensus vector W_k is calculated as the average of sign-corrected loading vectors in M_k , i.e., $W_k := \frac{\sum_{w_{ij} \in M_k} \tilde{w}_{ij}}{|M_k|}$. Note that the average is taken not over the original loading vectors w_{ij} , but rather their sign-corrected versions \tilde{w}_{ij} . Specifically, the signs of each w_{ij} in M_k are corrected so that all loading vectors have positive cosine coefficients.
 - a. *We note that, given a specific cosine threshold for constructing edges of the network, it is not guaranteed that such a correction is always possible. That is, with all possible sign corrections, there are still certain intracluster PCs that have negative cosine coefficients. Such cases are unlikely to happen in empirical evaluations, and are further reduced by our modularity clustering approach (**Supplemental Fig. 17**). We discuss this issue in **Supplemental Materials**.*
 - b. *In the case where such issues occur, a higher cosine threshold is recommended. With a sufficiently high cosine threshold, clusters are guaranteed to be consistent (all PCs will have positive cosines), but also be smaller and thus are less interpretable in terms of consistent biological signals across studies.”*

Line 385-386 (bullet point 2): Is there a default or guidance on how to choose the customizable similarity threshold?

Given our theoretical low bound empirical evidence, we recommend the default cosine cutoff of 0.7 (close to 0.717, which is the theoretical bound that guarantees all size-three clusters are consistent), but also suggest that the user vary this threshold slightly if needed to ensure robustness. We also implement error messages for the user to increase this threshold should inconsistent clusters be identified. In addition to the items above, point 2 is now also expanded to:

2. “For two PC loadings from different datasets w_{ij} and $w_{i'j'}$, similarity is quantified with the absolute value of cosine coefficient[19] $|\cos \langle w_{ij}, w_{i'j'} \rangle|$. This yields a network of PC loading vectors associated by weighted edges w_{ij} and $w_{i'j'}$, retaining edges only if their weight surpasses a customizable similarity threshold ($|\cos \langle w_{ij}, w_{i'j'} \rangle| > threshold_s, 0 < threshold_s < 1$).
 - a. This threshold is default to 0.7, which is close to the theoretical guarantee that all size-three clusters will by definition have positive cosine coefficients between all PC pairs. In practice, we recommend the user to vary this parameter to inspect for most interpretable results.”

Line 387: The reference for community detection is just to igragh, and it's not clear from the language here what exactly is happening. Is the reader meant to look at the previous paper for the details?

This is addressed above (expanded **Methods** and **Supplemental Materials** text).

Line 393-4 (point 4): It's not clear to me that it is always possible to have the signs work out. It will depend on the details of the thresholds for the similarity threshold and the denseness of the connectivity for the modules. Suppose we have the following three PCs:

[,1] [,2] [,3]

[1,] -1.38650440 -0.3022241 -1.268658799

[2,] 1.72311730 1.6626374 0.518501652

[3,] -0.64458549 -1.1139292 0.281648595

[4,] -0.42806325 -1.7377230 1.285245698

[5,] -0.06428050 -1.2537129 1.099791236

[6,] 1.09050667 1.5585543 -0.158618842

[7,] 0.61818275 -1.0355818 1.973135085

[8,] 1.49500607 0.9051074 0.990755655

[9,] -0.85899293 0.4665796 -1.610858682

[10,] 1.70777175 0.9993662 0.984633074

[11,] 0.94210852 1.1281060 -0.008446498

[12,] 1.05446578 1.0187603 0.467260791

[13,] -0.05649655 0.3119414 -0.388900395

[14,] 1.32647536 -0.1490256 1.716939688

[15,] -0.41847582 -0.4641675 -0.122296172

[16,] -2.02838181 -0.8189566 -1.433360942

[17,] 0.68963249 0.2148923 0.524455833

[18,] -0.80249485 -1.0057786 -0.069795483

[19,] -0.16633605 -0.5568348 0.425177277

[20,] -0.44020648 -0.5360575 0.194269493

The cosine similarity between 1 and 2 and 1 and 3 is about .65, but the cosine similarity between 2 and 3 is -.13 (if I did the computations correctly). This would lead to a network where $1 \leftrightarrow 2$, $1 \leftrightarrow 3$ if the cosine similarity threshold is .6, and I assume this would be a densely connected module if the threshold for the fraction of connections is .6 as well (because the module has $2/3 = .67$ possible connections). However, it is not possible to change all the signs in this example so that all pairs of cosine similarities are positive.

This phenomenon is why I asked about the details of the cluster detection and the threshold for the cosine similarity. If these cutoffs are stringent enough, you don't run into problems like this, but if they're a little looser you can.

This should also be addressed in the overview above (and especially new **Supplemental Fig. 17**, expanded **Methods** and **Supplemental Materials** text).

Line 395: It is not clear to me that the consensus vectors need to be mutually independent, even approximately. Consider the case where all of the studies have the same set of PCs, but the eigenvalues corresponding to the top PCs are the same and the PCs are not identifiable. Then even with infinite data within each study, the PC network will just be connecting random components from the principal plane and there is no guarantee that the consensus components will be orthogonal.

While we agree with this in the abstract, it is exceptionally unlikely to occur in real data. If we think of microbial data as being generated by low-dimensional biological/technical factors plus noise, then the scenario suggested by the reviewer would require all such factors to have exactly the same sized loading. This is a measure-zero set that we reason is highly unlikely to happen for real-world data. Thus at least asymptotically, the PCs should be identifiable.

It could be argued PCs can nevertheless be “near-unidentifiable”, that is, some top PCs may have eigenvalues that are close enough to cause identifiability issues in their spanned subspace in finite samples. In this case, we note that it is still extremely unlikely for random high-dimensional PCs to correlate, let alone to form clusters. This follows from the observation that the probability for two points uniformly distributed on the unit sphere to form a small angle (i.e., have a large cosine coefficient) decreases rapidly with the dimensionality of the sphere.

While this is well-established theoretically, we also simulated two 100-dimensional PCs over 10,000 iterations, both uniformly distributed on the 100-dimensional unit sphere. Out of the iterations, no absolute cosine of > 0.5 was observed. This shows that the scenario suggested by the reviewer is, again, at best highly improbable for our target application (high-dimensional microbiome data).

Line 400 (point 6): I think a better way to do this would be to validate by checking the proportion of variance explained by the identified component in the new dataset instead of correlating the identified component with the top principal components in the new dataset. There are two reasons for this: 1 - If the eigenvalues for the top PCs are similar, there is a lack of identifiability in the actual PCs, and you are stacking the deck against yourself by requiring the PCs to correlate exactly. 2 - Even if you don't have the issue with similar eigenvalues/non-identifiability of the PCs, it seems that it should be "good enough" for the component you identified in the training data to explain a lot of variance in the test set. It seems a bit too strict a criterion for the component to have to correlate with the top PC in the test set. If you're getting that sort of correlation anyway it's probably not worth it to change to a new metric, but I think a less stringent criterion is justified.

This is a very interesting alternative approach, and we agree with the reviewer that it could provide benefits to this sub-component of MMUPHin under certain circumstances. However, as illustrated in the above response, the identifiability concern does not hold for real-world applications of the methodology. A similar percentage variability should be good evidence that continuous score is validated in the dataset, though. We would be happy to implement and evaluate this as an alternative means of continuous pattern discovery in future updates to MMUPHin, although we will retain the published approach in the current version.

Reviewer #3

This ms tried to set up a statistical framework for meta-analysis of datasets from various independent studies. Such attempt is urgently needed for the microbiome field. However, the taxon-based approach employed in this ms represents a huge concern that has hampered the progress of human microbiome field for too long.

The first concern is the ignoring of novel bacteria by the taxon-based analysis which is database-dependent. Sequences representing novel members of the gut microbiome will be left aside from any downstream analysis because they have no nearest neighbor in the database and cannot be given a proper taxonomic position. Without a taxonomic name, such bacteria cannot be lumped together with other bacteria having the same name to reduce the dimensionality of the microbiome datasets. If we only focus on bacteria which have been previously characterized but ignore novel ones, we may not move forward on new findings on gut microbiome in human health and diseases.

We wholeheartedly agree with the reviewer that novel methods are important in the study of microbial communities, particularly for uncharacterized sequences and taxa; we and others have reviewed this topic previously[20-22]. We also agree that many other types of microbiome study can be facilitated using MMUPHin; that is, it is not limited to analysis of OTUs or even of amplicon data. The model and implementation are both generalizable to different types of taxonomic features (e.g. ASVs, species, MAGs, or SGBs) or functional features (e.g. genes, pathways, or contigs). We have included an additional validation of MMUPHin's results on ASVs from the currently-meta-analyzed 16S data above in response to Reviewer #1, as well as finding no substantial difference in results for either type of sequence processing (New Supplemental Figure 18).

We are, additionally, currently carrying out an additional meta-analysis using over 2,300 shotgun metagenomes from over 500 additional IBD study participants, as an extension of this initial work using a larger set of amplicon profiles. Since, even now, microbiome publications using amplicon data represent approximately an order of magnitude greater sample numbers than those using shotgun metagenomes, however, we thought it best to initially validate novel methodology for combining microbiome datasets using the more widespread data type.

Importantly, we should also point out that the reviewer may misunderstand some of the strengths and limitations of various sequence-based microbial community analysis methods. Taxon-based analyses (including ours) do not necessarily ignore novel bacteria, as amplicon sequences of unknown or uncertain taxonomy can be very reliably assigned to ASVs or OTUs[1, 23]. The same is true for shotgun metagenomic taxonomy via combinations of reference- and assembly-based methods [22, 24]. Taxon-based analysis is also not intrinsically database-dependent, although it can be if misapplied; the same sequence can be assigned two different "names" by different databases, but this will not affect analyses based either on the sequence identifier itself, or on its phylogenetic placement[25]. It is never the case that a sequence has "no nearest neighbor in the database" - at worst, its phylogenetic placement will be

highly uncertain (e.g. “Bacterial”) - and such sequences, again, are neither omitted from further analysis nor “lumped together” (that is, it is entirely possible - and common - for two or more groups of different, unnamed sequences to be analyzed individually). We thus assure the reviewer that neither this analysis, nor our previous studies, ignore uncharacterized microbes - while, again, enthusiastically agreeing that more work must be done to better-understand them.

Finally, we note that one of the biggest strengths of meta-analysis is to synthesize consistent but weakly-powered associations across individual studies and report confident findings[26]. As has been very much the case in e.g. meta-analysis of genome-wide association studies into very large, joint populations, this permits novel discoveries even within known features that were previously too weak to detect. In our context of interest, this is exemplified by our reports on *Turicibacter* and *Acinetobacter*, which have not previously been studied in individual investigations of the IBD microbiome.

The second concern is that after removing unclassified bacteria from the datasets, how much of the left datasets can still reflect the true quantitative relationships of all the known bacteria? If the left dataset cannot reflect the ecological structure of the gut microbiome, any findings from such analysis may be spurious and non-reproducible.

As above, we again clarify that bacteria that are not fully classified were not removed in our analysis. That is, OTUs that were only resolved at e.g. the family level were still included and reported as such (e.g. feature *Escherichia_unclassified*). This ensures that unique sequences, even if not yet named, are analyzed as such. Further, it is unclear how omission of unclassified sequences would make findings non-reproducible; in our work, OTU profiles were generated with the same bioinformatics pipeline, but independently so for each dataset. Thus, if these profiles indeed do not reflect true gut microbiome structures, findings from different studies should be inconsistent and would not be successfully synthesized with meta-analysis. This is obviously not true, given that our identified associations are recurrent across studies, by the very definition of meta-analytical findings (**Supplemental Table 6**).

The third concern is related with using taxa as a functional unit for association studies. Bacterial species is technically defined as strains which share higher than 70% homology (DNA-DNA hybridization) or higher than 95% ANI (genome sequence comparison). This means that members in the same bacterial species can have up to 30% difference in their genomic sequences. Because of this, members in the same species can change in opposite directions when disease progresses. This means that some bacteria may increase, others may decrease when the disease gets worse or improved, though all these bacteria may belong to the same species. When you move up to genus level, the situation become even worse as members in the same genus can have only 25% genome homology. If bacteria in the same species/genus or any other taxon level change in opposite directions, any attempts in lumping them together will lead to a new variable which no longer reflect any real ecological relationships among individual bacteria. Unfortunately, this ms started to establish the new statistical framework by looking at phylum level variations. Even species is not a unit with enough resolution for functionally dissecting bacteria, going to phylum

level is indeed a huge problem. This is why phylum level microbiome signatures have been the least reproducible. Genus level signatures are also very difficult to be reproducible.

This ms used OTUs as the first level organization for data analysis. The resolution of OTUs is roughly at species level. This is not enough. OTUs do not take advantage of the possible subspecies level resolution that the sequences can offer. The ms should use ASVs as the basic units for analysis.

There's also a widely held misunderstanding on 16S rRNA gene sequencing. Many people believe that you can only go down to genus level when you analyze amplicon sequences. This is only true if you want to do taxon-based analysis. Each ASV represent a unique group of bacteria whose genome homology is somewhere between individual strains and species. You don't need a name to be able to follow the behavior of a bacterium. Each AVS with a unique ID represents a unique group of bacteria. By examining the ecological behavior of ASVs, you will be able to identify unique groups of bacteria which are correlated with host phenotypes. the resolution will be the much higher than any database-dependent and taxon-based analysis.

As above, we appreciate the reviewer's concerns, and agree that there are both many appropriate sequence-based assays for studying microbial communities, and many different ways to produce and analyze amplicon-based sequences. The reviewer may be mistaken about the relationship between the technologies and bioinformatics approaches they discuss, however, and the material being studied in this work - specifically, the MMUPHIn model for microbiome meta-analysis and its application to IBD. Neither our own study, nor most of the individual prior publications integrated into the meta-analysis, are dependent on the exact definitions of microbial features used - sequence similarity thresholds, taxonomy assignment, or even that the features represent taxa at all. The 16S sequences used from the RISK cohort, for example [15], were originally analyzed using as closed-reference OTUs using QIIME 1.7, predating the availability of ASV-based approaches; those in PROTECT[27] using UPARSE 8.1; and those in the HMP2 [8] using dada2. We have uniformly re-processed them using established methodology [24], but this is neither an aspect of MMUPHIn, nor does it have any influence on our biological findings in IBD. Indeed, as explained above, part of the strength of MMUPHIn's meta-analysis component is that only results that are consistent among studies will remain significant, by definition.

We would also point out several slight oversimplifications in the reviewer's assumptions that may impede their interpretation of our study. No single definition of microbial species is, to our knowledge, universally accepted, although 95% ANI is generally an excellent approximation[28]. However, this percent identity applies to whole-genome sequences, neither to the full-length 16S rRNA gene, nor to the V4 hypervariable region as most commonly amplified in the studies integrated in our meta-analysis[29]. It is also unclear why the reviewer claims that phylum-level analyses were used in our studies, as these do not occur at any point. Instead, individual studies were re-analyzed to OTUs as previously described, and meta-analyzed between studies using genus-level taxonomy as the most consistent inter-study feature type; as discussed below, this is not a limitation of our method, but instead the most appropriate approach when comparing differing 16S data generation (not bioinformatic)

protocols[30]. Finally, both OTUs and ASVs allow analysis of microbial taxonomic units with or without nomenclature, with comparable resolution (within the limits of the primers and hypervariable regions chosen for data generation) *as long as* appropriate denoising is carried out[31].

We agree that it would, in principle, be possible to re-analyze the entire meta-analysis dataset using a different approach, such as varying OTU percent identity thresholds, quality control measures, or using ASVs instead, but doing so would entail essentially an additional new manuscript. While we would be interested in pursuing such further validation work in the future, we do not feel that it would be appropriate for this study.

Instead, we have already begun work on a follow-up investigation applying MMUPHin to a similar meta-analysis of taxonomic and functional features from additional IBD gut shotgun metagenomes. While a full preprint is not quite advanced enough for supplementary inclusion, briefly, we applied the bioBakery workflows (for bioinformatic processing) and MMUPHin (for meta-analysis) to seven previously published investigations characterizing gut microbial ecology in IBD, encompassing 2,371 metagenomic samples from 542 unique participants (**Response Fig. 2A**). Study cohorts enrolled participants from across the United States and Europe, and each utilized non-uniform sampling strategies based on fundamental study design decisions (e.g., cross-sectional vs. longitudinal), disease status (inception vs. established disease), target age of the study population (adult, pediatric, or both), and treatment status (naive vs. experienced), among others (**Response Fig. 2B**).

Response Figure 2: Preliminary meta-analysis of IBD shotgun metagenomes using MMUPHin. **a)** Baseline characteristics of meta-analyzed study participants by age cohort and disease. **b)** Participant disease status, sex, and prior antibiotic usage by study cohort. **c)** Prior to meta-analysis, species median abundance by cohort showing top 25 universal (i.e., found in all seven cohorts), overlapping (more than 1), and solo (found in just one) taxa ranked by median abundance. **d)** Principal coordinate analysis plots using species-level Bray-Curtis distances show broad shifts in population structure attributable to disease status (left) with characteristic gradation of the Bacteroidetes and Firmicutes phyla along major axes of variation (below). Abbreviations: CD (Crohn’s disease), HC (healthy control or non-IBD), PCoA (principal coordinates analysis), UC (ulcerative colitis).

To identify harmonized species-resolved signatures of IBD from these metagenomes, analogous to our 16S genera, we performed a feature-level meta-analysis using MMUPHin’s batch correction and linear

mixed effects. We modeled the jointly normalized log-transformed taxa abundances against disease status after accounting for age, sex, prior antibiotic use, and random cohort/subject effects. Differential abundance testing was conducted with comparisons between IBD, CD, or UC vs. non-IBD and between IBD subtypes (i.e., CD vs. UC). Across these 4 comparisons, in total, we identified 84 differentially abundant species among 160 statistically significant association tests after correction for multiple hypothesis testing (FDR<0.05). Compared to non-IBD gut microbiomes, we observed a relative enrichment of *Ruminococcus gnavus* and several *Clostridium spp.*, whereas *Subdoligranulum spp.*, *Alistipes putredinis*, and *Eubacterium rectale* were depleted (**Response Fig. 3**).

Response Figure 3: Shared and distinguishing taxonomic signatures of IBD gut metagenomes. Left panel: meta-analysis summary statistics (using β -coefficient and SE from linear mixed effects modeling) from aggregated study samples. Point estimates are colored by FDR-corrected p-value (main panel) with results for IBD compared to non-IBD (above) and the features able to discriminate CD from UC (below). Disaggregated (i.e., single study) results (right) demonstrates broad agreement in the effect size and direction of microbial alterations in IBD and its subtypes, while highlighting the power of a meta-analytic approach to identify consensus microbial alterations not observed in single study cohorts alone. Inset circles represent study weight contributed to the aggregated meta-analysis, and asterisks (*) signify level of statistical significance of linear association testing within a given cohort (* = FDR<0.1, ** = FDR<0.05). Another heatmap (far right panel) displays the meta-association testing results for species abundance and confounding factors (age, sex, and prior antibiotic use). Abbreviations: CD (Crohn's disease), FDR (false discovery rate), IBD (inflammatory bowel disease), UC (ulcerative colitis).

Next, using a methodological approach similar to our taxonomic analysis, we characterized differences in functional pathways encoded by gut microbial communities in IBD. As in both of our taxonomic analyses, while individual studies demonstrated broad concordance in the pathways differentially associated with disease status, the statistical power to detect these relationships could only be achieved

after meta-analysis. In particular, we observed a novel augmentation in microbial molybdenum cofactor biosynthesis (PWY-6823), contributed primarily by *Escherichia coli* (**Response Fig. 4**). This is a possible consequence of greater microbial utilization of molybdenum-containing enzymes, such as nitrate reductase, that are typically advantageous in the volatile milieu of the IBD gut, an environment characterized by increases in both nitric oxide and DUOX2. Prior experimental models of IBD have shown that *E. coli* deficient for molybdenum cofactor biosynthesis were comparatively less viable than wild-type[32].

Additionally, we found significant enrichment of heme acquisition/biosynthesis pathways in IBD, suggesting that specialized functions related to the use of iron and other metals may be context (i.e., disease) dependent. Further, it is well known that fecal lactoferrin, an iron-binding glycoprotein released by activated neutrophils, can serve as a clinically measurable biomarker for intestinal inflammation. Taken together, enrichment of this biosynthetic pathway could confer a fitness advantage in the IBD gut where iron may be comparatively scarce, and could, in part, help explain the significant expansion of *Escherichia* and other Proteobacteria in IBD.

Response Figure 4: Pathways related to saccharide degradation and B12 biosynthesis altered in IBD. MetaCyc pathway summary statistics (in relative abundance and 95% CI) from meta-analyzed linear models. Point estimates colored by FDR-corrected p-value (main panel) with results for IBD compared to non-IBD (above) and the CD vs. UC (below). Study-level heatmap (right) demonstrates broad agreement in the effect size and direction, but no statistically-significant within-study differences. Inset circles represent study weight contributed to the aggregated meta-analysis, and asterisks (*) signify level of statistical significance of linear association testing within a given cohort (* = FDR<0.1, ** = FDR<0.05). Barplot (far right) indicates top 15 most differentially abundant taxa contributing to disrupted pathways, presented as the difference in mean abundance between cases vs. reference (e.g., IBD vs. non-IBD or CD vs. UC). Abbreviations: CD (Crohn's disease), CDUC (Crohn's disease vs. ulcerative colitis), FDR (false discovery rate), IBD (inflammatory bowel disease), UC (ulcerative colitis).

As is probably already apparent from these few examples, there is more than enough material to focus on other types of biological features in the IBD microbiome during future analyses, including strains, microbial genetics, and functional genes and pathways. We also hope that this is ample evidence, however, that the MMUPHin methodology itself is generalizable to these other settings (since all of these results employ methods near-identical to the current manuscript, save for the underlying data types).

THE HARVARD CHAN
MICROBIOME IN
PUBLIC HEALTH CENTER

Finally, these examples also speak to our awareness of and excitement about continuing to work with other types of microbial feature definitions for such analyses in the future, as suggested by the reviewer.

References

1. Callahan BJ, McMurdie PJ, Rosen MJ, Han AW, Johnson AJ, Holmes SP: **DADA2: High-resolution sample inference from Illumina amplicon data.** *Nat Methods* 2016, **13**:581-583.
2. Armour CR, Topcuoglu BD, Garretto A, Schloss PD: **A Goldilocks Principle for the Gut Microbiome: Taxonomic Resolution Matters for Microbiome-Based Classification of Colorectal Cancer.** *mBio* 2022:e0316121.
3. Nazarieh M, Rajula HSR, Helms V: **Topology Consistency of Disease-specific Differential Co-regulatory Networks.** *BMC Bioinformatics* 2019, **20**:550.
4. Gibbons SM, Duvallet C, Alm EJ: **Correcting for batch effects in case-control microbiome studies.** *PLoS Comput Biol* 2018, **14**:e1006102.
5. Mallick H, Rahnavard A, McIver LJ, Ma S, Zhang Y, Nguyen LH, Tickle TL, Weingart G, Ren B, Schwager EH, et al: **Multivariable association discovery in population-scale meta-omics studies.** *PLoS Comput Biol* 2021, **17**:e1009442.
6. Bogart E, Creswell R, Gerber GK: **MITRE: inferring features from microbiota time-series data linked to host status.** *Genome Biol* 2019, **20**:186.
7. Hawinkel S, Mattiello F, Bijnens L, Thas O: **A broken promise: microbiome differential abundance methods do not control the false discovery rate.** *Brief Bioinform* 2019, **20**:210-221.
8. Lloyd-Price J, Arze C, Ananthakrishnan AN, Schirmer M, Avila-Pacheco J, Poon TW, Andrews E, Ajami NJ, Bonham KS, Brislawn CJ, et al: **Multi-omics of the gut microbial ecosystem in inflammatory bowel diseases.** *Nature* 2019, **569**:655-662.
9. Wirbel J, Zych K, Essex M, Karcher N, Kartal E, Salazar G, Bork P, Sunagawa S, Zeller G: **Microbiome meta-analysis and cross-disease comparison enabled by the SIAMCAT machine learning toolbox.** *Genome Biol* 2021, **22**:93.
10. Zhang Y, Parmigiani G, Johnson WE: **ComBat-seq: batch effect adjustment for RNA-seq count data.** *NAR Genom Bioinform* 2020, **2**:lqaa078.
11. Tran HTN, Ang KS, Chevrier M, Zhang X, Lee NYS, Goh M, Chen J: **A benchmark of batch-effect correction methods for single-cell RNA sequencing data.** *Genome Biol* 2020, **21**:12.
12. Ma S, Ren B, Mallick H, Moon YS, Schwager E, Maharjan S, Tickle TL, Lu Y, Carmody RN, Franzosa EA, et al: **A statistical model for describing and simulating microbial community profiles.** *PLoS Comput Biol* 2021, **17**:e1008913.
13. Morgan XC, Tickle TL, Sokol H, Gevers D, Devaney KL, Ward DV, Reyes JA, Shah SA, LeLeiko N, Snapper SB, et al: **Dysfunction of the intestinal microbiome in inflammatory bowel disease and treatment.** *Genome Biol* 2012, **13**:R79.
14. Qin J, Li Y, Cai Z, Li S, Zhu J, Zhang F, Liang S, Zhang W, Guan Y, Shen D, et al: **A metagenome-wide association study of gut microbiota in type 2 diabetes.** *Nature* 2012, **490**:55-60.
15. Gevers D, Kugathasan S, Denson LA, Vazquez-Baeza Y, Van Treuren W, Ren B, Schwager E, Knights D, Song SJ, Yassour M, et al: **The treatment-naive microbiome in new-onset Crohn's disease.** *Cell Host Microbe* 2014, **15**:382-392.
16. Ma S, Ogino S, Parsana P, Nishihara R, Qian Z, Shen J, Mima K, Masugi Y, Cao Y, Nowak JA, et al: **Continuity of transcriptomes among colorectal cancer subtypes based on meta-analysis.** *Genome Biol* 2018, **19**:142.
17. Brandes U, Delling D, Gaertler M, Gorke R, Hofer M, Nikoloski Z, Wagner D: **On modularity clustering.** *IEEE transactions on knowledge and data engineering* 2007, **20**:172-188.

18. Csardi G, Nepusz T: **The igraph software package for complex network research.** *InterJournal* 2006, **Complex Systems**:1695-1695.
19. Jaskowiak PA, Campello RJ, Costa IG: **On the selection of appropriate distances for gene expression data clustering.** *BMC Bioinformatics* 2014, **15 Suppl 2**:S2.
20. Lloyd-Price J, Abu-Ali G, Huttenhower C: **The healthy human microbiome.** *Genome Med* 2016, **8**:51.
21. Joice R, Yasuda K, Shafquat A, Morgan XC, Huttenhower C: **Determining microbial products and identifying molecular targets in the human microbiome.** *Cell Metab* 2014, **20**:731-741.
22. Pasolli E, Asnicar F, Manara S, Zolfo M, Karcher N, Armanini F, Beghini F, Manghi P, Tett A, Ghensi P, et al: **Extensive Unexplored Human Microbiome Diversity Revealed by Over 150,000 Genomes from Metagenomes Spanning Age, Geography, and Lifestyle.** *Cell* 2019, **176**:649-662 e620.
23. Edgar RC: **UPARSE: highly accurate OTU sequences from microbial amplicon reads.** *Nat Methods* 2013, **10**:996-998.
24. Beghini F, Mclver LJ, Blanco-Miguez A, Dubois L, Asnicar F, Maharjan S, Mailyan A, Manghi P, Scholz M, Thomas AM, et al: **Integrating taxonomic, functional, and strain-level profiling of diverse microbial communities with bioBakery 3.** *Elife* 2021, **10**.
25. Hamady M, Knight R: **Microbial community profiling for human microbiome projects: Tools, techniques, and challenges.** *Genome Res* 2009, **19**:1141-1152.
26. Viechtbauer W: **Conducting Meta-Analyses in R with the metafor Package.** *2010* 2010, **36**:48.
27. Hyams JS, Davis Thomas S, Gotman N, Haberman Y, Karns R, Schirmer M, Mo A, Mack DR, Boyle B, Griffiths AM, et al: **Clinical and biological predictors of response to standardised paediatric colitis therapy (PROTECT): a multicentre inception cohort study.** *Lancet* 2019, **393**:1708-1720.
28. Jain C, Rodriguez RL, Phillippy AM, Konstantinidis KT, Aluru S: **High throughput ANI analysis of 90K prokaryotic genomes reveals clear species boundaries.** *Nat Commun* 2018, **9**:5114.
29. Soergel DA, Dey N, Knight R, Brenner SE: **Selection of primers for optimal taxonomic classification of environmental 16S rRNA gene sequences.** *ISME J* 2012, **6**:1440-1444.
30. Sinha R, Abu-Ali G, Vogtmann E, Fodor AA, Ren B, Amir A, Schwager E, Crabtree J, Ma S, Microbiome Quality Control Project C, et al: **Assessment of variation in microbial community amplicon sequencing by the Microbiome Quality Control (MBQC) project consortium.** *Nat Biotechnol* 2017, **35**:1077-1086.
31. Prodan A, Tremaroli V, Brolin H, Zwinderman AH, Nieuwdorp M, Levin E: **Comparing bioinformatic pipelines for microbial 16S rRNA amplicon sequencing.** *PLoS One* 2020, **15**:e0227434.
32. Hughes ER, Winter MG, Duerkop BA, Spiga L, Furtado de Carvalho T, Zhu W, Gillis CC, Buttner L, Smoot MP, Behrendt CL, et al: **Microbial Respiration and Formate Oxidation as Metabolic Signatures of Inflammation-Associated Dysbiosis.** *Cell Host Microbe* 2017, **21**:208-219.

Second round of review

Reviewer 1

The authors have clearly devoted a large amount of time and effort to their revisions of this manuscript, and I particularly appreciate the way in which they organized their responses to each individual aspect of each review.

In the expanded discussion of the analysis of 16S ASVs, the authors do not consider the potential confounding effects of the primers and amplification protocols used across studies. Rather than viewing this as a deficiency of their manuscript, I have come to see it instead as being out of their scope of effort. In fact, their emphasis on the utility of making comparisons at the genus level may be seen as a practical way of limiting the confounding effects of variable taxonomic resolution across amplicons. It would likely be out of scope for this tool to practically combine 16S data generated with different variable regions, based on my reading of the paper.

Overall, I would say that the authors have very appropriately and satisfactorily responded to the questions and comments made by all reviewers.

Reviewer 2

My previous concerns were addressed in the revised manuscript. I have two major comments. The first is a follow-up question related to the effectiveness of the batch correction that should be simple for the authors to address.

If I understand correctly the relationship between the components of MMUPHin, the batch correction is not expected to completely get rid of batch or study effects, but to mitigate them so that the meta-analysis components have more power. My question has to do with how important the batch correction part is. If the meta-analysis component was applied to the uncorrected profiles, how well would it perform? In particular, can Figure 2c be supplemented to include such a scheme? (I wasn't entirely sure what procedure (a) in the paragraph beginning on line 515 meant, I believe it means that MaAsLin2 is applied to all of the uncorrected data as if it came from the same study, which is different from what I am proposing, but if my understanding of (a) is incorrect the authors should clarify in the manuscript.)

My second major comment is that parts of the paper, the Results section in particular, are very hard to follow and would benefit greatly from editing for flow. If the reader works hard enough, he can figure out what the authors were thinking, but this kind of writing is not likely to make the reader positively disposed to the work. The authors should consider taking some of the standard advice about how to structure papers, paragraphs, and sentences. A good reference is "Ten simple rules for structuring papers," by Mensh and Kording in PLoS CB (<https://doi.org/10.1371/journal.pcbi.1005619>).

For example the first paragraph in "A statistical framework for meta-analysis of microbial community profiles" reads:

> "It consists of three main components: batch and study effect correction, covariate modeling, and population structure discovery. First, we extended methods from the gene expression

literature (ComBat[15]) to enable batch correction of zero-inflated microbial abundance data. Based on linear modeling, the method can differentiate between technical effects (batch, study) versus covariates of biological interest (exposure, phenotype). Second, we combined well-validated data transformation and linear modeling combinations for microbial community profiles[33] with fixed and random effect modeling[34] for meta-analytical synthesis of per-feature (taxon, gene, or pathway) differential abundance effects. Lastly, we generalized and formalized approaches from cancer transcriptional subtyping[35] to permit unsupervised discovery and validation of both discrete and continuous population structures in microbial community data.”

In this paragraph, I want to know what the method does and why (the “context” and “content” from the PLoS CB paper referred to above). How it is done is of secondary importance, as it will be described in detail in the “Methods” section. The first sentence is good. In the second sentence, I am expecting the context and content for the first component of MMUPHIn. Instead, the sentence starts off with the implementation details (“based on linear modeling”). I might care about that information later, but in this context it’s something that I have to carry around in my head while I wait for the context and content. Same goes for sentences three and four (“we combined well-validated data transformation and linear modeling combinations...” and “we generalized and formalized approaches from cancer transcriptional subtyping”), which is again telling us about implementation details when what we want are the context and the content. A version of this paragraph that would be less frustrating to the reader would be something like:

> "MMUPHIn consists of three main components: batch and study effect correction, covariate modeling, and population structure discovery. To correct for batch and study effects, the first component of MMUPHIn extends methods from the gene expression literature to give the user batch-corrected microbial abundance profiles. Second, to test for differential abundance while taking into account study and batch effects that remain after batch adjustment, we apply fixed- and random-effects models developed for meta analysis. Finally, to permit unsupervised discovery of discrete and continuous population structures, we generalize approaches from cancer transcriptional subtyping."

This has all the same information, but within each sentence the context comes first. This makes it easier for the reader to follow the logic: each sentence starts out with a problem and then describes the solution. Contrast with the initial version, in which the implementation details are described first. In that case, the reader is left with a lot of unresolved questions (why did you extend methods from the gene expression literature? Would something else be better?) that are not answered until much later.

A large number of paragraphs in the Results section suffer from similar issues.

A related point that makes some of the paper difficult to follow is the fact that MMUPHIn has many components, but the authors seldom specify which part of the “collection of tools” they are referring to when they refer to MMUPHIn. Again, this is something that the reader can figure out if he works hard enough, but the lack of precision is not very considerate. In particular, I often found myself spending time trying to figure out whether the authors were referring to results based solely on batch-corrected profiles or to results based on meta-analysis of batch corrected

profiles (e.g. first paragraph of the section “Meta-analysis of the IBD microbiome”, the paragraph starting on line 115, the paragraph starting on line 126). As with many of my other comments, this is something the reader can figure out if he works hard enough, but it would be better for the authors to specify.

Some other notes:

- Line 91: This section is called “A statistical framework...” but it combines discussion of the statistical framework with discussion of the simulation studies. Discussion of the simulation should be in a separate section.
- Line 95: “Covariate modeling” is very jargony.
- Line 126: This paragraph discusses both differential abundance testing and structure discovery. They should be split up unless you want the reader to spend time trying to figure out what the relationship between differential abundance testing and population structure discovery is.
- Line 144: My understanding is that this paragraph discusses what the batch-adjusted profiles look like and that the results are not based on the meta-analysis part of the pipeline. If this is not true the authors should clarify. If this is true, I am a little bit confused about why these profiles are being discussed in such detail. There are still substantial batch effects at this point that have not been dealt with, and so while there is some interest in the effect of the batch adjustment, we can’t draw biological conclusions based on any of the phenomena described here.
- Line 360: This paragraph is confusing because it says the same thing twice. The “In practice” is supposed to signal that the authors are starting the explanation over again from the beginning, but on the first read I thought that the sentence was expanding on just the metaforR part of the pipeline, not the whole thing. I’m sure the authors can come up with something better, but I would suggest instead of “In practice”, something like “Overall, for the meta-analysis part of the pipeline, the user provides...”
- Line 525: Is this paragraph complete? Did the authors want to describe the results of the computational cost study?

Authors’ response to reviewers:

Reviewer 1

The authors have clearly devoted a large amount of time and effort to their revisions of this manuscript, and I particularly appreciate the way in which they organized their responses to each individual aspect of each review.

In the expanded discussion of the analysis of 16S ASVs, the authors do not consider the potential confounding effects of the primers and amplification protocols used across studies. Rather than viewing this as a deficiency of their manuscript, I have come to see it instead as being out of their scope of effort. In fact, their emphasis on the utility of making comparisons at the genus level may

be seen as a practical way of limiting the confounding effects of variable taxonomic resolution across amplicons. It would likely be out of scope for this tool to practically combine 16S data generated with different variable regions, based on my reading of the paper.

Overall, I would say that the authors have very appropriately and satisfactorily responded to the questions and comments made by all reviewers.

We thank the reviewer for their encouraging words. Regarding the technical effects of primers and amplification protocols, we would like to note that while MMUPHIn is agnostic towards primers, extraction protocols, amplicon regions, etc., it will nevertheless attempt to correct for study differences caused by such technical variables, to the extent that they are visible as batch effects. However, while we have not evaluated it quantitatively ourselves, it is likely that primer or variable-region protocol differences are large enough as to be incompletely eliminated at best (e.g. 32788589, 30834331, 30720800, 33221964, and many earlier studies).

The lesser, but still substantial, protocol differences between many 16S-based studies was indeed what motivated our genus-level approach to the analysis. We agree with the reviewer that in practice, to the extent possible, this approximation has reasonable cost-benefit in alleviating the effect of different amplicon regions. The following clarifying paragraph has been updated to the Methods section (subsection “Batch adjustment: MMUPHIn_Correct”):

“Lastly, we note that MMUPHIn_Correct does not explicitly model any particular sources of batch effects, such as primers, extraction protocols, and amplicon regions for 16S rRNA sequenced profiles. However, it will nevertheless attempt to correct for variability caused by differences in these protocols, to the extent that they manifest as batch/study differences. As examples: if two studies adopted different extraction protocols, potential study differences will be captured with MMUPHIn_Correct and normalized. In contrast, if samples within the same study were sequenced using different amplicon regions, and this difference in protocol was not flagged as a “batch” variable, MMUPHIn_Correct will not register the potential differences.”

Reviewer 2

My previous concerns were addressed in the revised manuscript. I have two major comments. The first is a follow-up question related to the effectiveness of the batch correction that should be simple for the authors to address.

If I understand correctly the relationship between the components of MMUPHin, the batch correction is not expected to completely get rid of batch or study effects, but to mitigate them so that the meta-analysis components have more power. My question has to do with how important the batch correction part is. If the meta-analysis component was applied to the uncorrected profiles, how well would it perform? In particular, can Figure 2c be supplemented to include such a scheme? (I wasn't entirely sure what procedure (a) in the paragraph beginning on line 515 meant, I believe it means that MaAsLin2 is applied to all of the uncorrected data as if it came from the same study, which is different from what I am proposing, but if my understanding of (a) is incorrect the authors should clarify in the manuscript.)

This is an important comment regarding the best approach to account for batch effects in differential abundance analysis generally, and it led us to investigate several aspects of this step in MMUPHin as they relate to previous work. Our overall response is: A) batch correction is usually not required to improve power for differential abundance meta-analysis; this is not because MMUPHin leaves residual batch effects, but rather a universal property for supervised testing in the context of batch effect adjustment. B) Instead, the effectiveness of MMUPHin batch correction is best evaluated through the overall reduction of batch differences in the microbial profiles (**Fig. 2a**), and its benefit is most obvious in visualization or unsupervised analysis where adjusting for confounding batch effects is not immediately straightforward (**Fig. 2b, e-h**). We detail the two parts below.

First, we note that batch correction before meta-analysis is not meant to improve power, supported by both theoretical and simulation evidence. Intuitively, *as long as uncertainty in the data is properly propagated*, differential abundance testing should have equivalent performance whether batch effect is adjusted for via joint modeling or in a two-stage fashion (where batch effects are partially/fully removed first).

To approach this theoretically, we consider the simple case of an exposure variable X_1 and a batch variable X_2 . We assume the true model is

$$Y = \beta_1 X_1 + \beta_2 X_2 + \epsilon$$

That is, the (transformed) microbial abundance Y is affected by both the exposure and batch effects through a linear model, with an independent error term ϵ . Here, the primary interest is to estimate the exposure differential abundance effect β_1 .

If we fit a linear model, adjusting for batch effects, directly on the original abundance Y :

$$Y \sim X_1 + X_2$$

This corresponds to differential abundance testing, accounting for batch effects, but without correcting the data first (the approach suggested by the reviewer). By linear algebra, we can obtain that the variance for the estimator of β_1 obtained this way, $\widehat{\beta}_1^{No\ correction}$, is $[\sigma^2(X'X)^{-1}]_{(1,1)}$, where $X = (X_1, X_2)$ is the design matrix, and σ^2 is the variance of ϵ .

Alternatively, we can first batch correct the data, and then fit the above model (the approach evaluated in the original **Figure 2c**). If we perfectly estimate and account for batch effects, the adjusted microbial abundances would be $\tilde{Y} = Y - \beta_2 X_2 = \beta_1 X_1 + \epsilon$. We now fit the linear model on the corrected data:

$$\tilde{Y} \sim X_1 + X_2$$

By linear algebra, we can derive that the variance for β_1 's estimator, $\widehat{\beta}_1^{Correction}$, is still $[\sigma^2(X'X)^{-1}]_{(1,1)}$, simply because the variance of \tilde{Y} , conditional on X , is the same as Y .

This means $\widehat{\beta}_1^{Correction}$ and $\widehat{\beta}_1^{No\ correction}$ have no difference in terms of efficiency, i.e., power. This conclusion is derived for the extreme case where batch effects are perfectly accounted for with oracle knowledge, but should be generalizable to other real-world implementations with any degree of conservativeness (including MMUPHin), as long as the variability of the batch correction stage is properly propagated to the batch-adjusted testing stage.

We followed up on this with empirical evidence through simulation evaluation, as requested by the reviewer. Specifically, we simulated both batch effects (effect size 2) and exposure effects (varying from 0 to 5) under batch imbalance of exposures at 40%, sample size per batch at 500, total 200 features, and 5% spiked-in features. We note that with or without batch-correcting the data first, MMUPHin's meta-analysis performed very similarly in terms of power and false positive rates.

New Figure S8: MMUPHIn meta-analyzed differential abundance testing has robust and consistent performance with or without upstream batch correction. For this simulation, we fixed the effect of the simulated batch variable at 2, while varying the effect of the simulated exposure variable. MMUPHIn_MetaDA yielded consistently valid performance (i.e., controlled false positive rates) with or without correcting the data first with MMUPHIn_Correct.

To conclude this first response point, we note that the necessity of correcting for batch effects prior to supervised meta-analysis is a widely discussed issue. In fact, as suggested by one of the co-authors on the original ComBat paper, batch correction is often not required as long as the batch variable is adjusted for in differential abundance analysis (<https://support.bioconductor.org/p/72815/>). However, it can still be useful, especially because methods such as ComBat and MMUPHIn adjust for both location and scale batch effects, while typical linear modeling based meta-analysis differential abundance testing only accounts for the former.

Second – and most importantly – the usefulness of MMUPHIn batch correction is best exemplified not with supervised meta-analysis, but rather in tasks for which adjusting for batch effects intrinsically would not be straightforward. Such tasks include e.g. visualization (**Figure 2b,f,h**), unsupervised clustering, and continuous score discovery (**Figure 2e,g**). This is consistent with the most common use cases of existing batch-correction methods in e.g. gene expression literature [1, 2].

To reflect these considerations, we have added the following text to **Methods** with the accompanying new **Figure S8**:

“We note that MMUPHIn_MetaDA always accounts for the batch variable in its supervised differential abundance testing. This agrees with the field’s consensus on the most appropriate way to address batch effects during supervised testing[1, 2]. Through simulation evaluations, the performance (FPR, power) of MMUPHIn_MetaDA is robust with or without upstream adjustment with MMUPHIn_Correct (**Additional File 1: Fig. S8**). Nevertheless, pre-correcting the data with MMUPHIn_Correct can still be helpful. This is both consistent with similar applications of batch correction in other molecular data types[1], and because MMUPHIn_Correct accounts for both location and scale batch effects, while the linear modeling in MMUPHIn_MetaDA only accounts for the former. Regardless, correcting the data with MMUPHIn_Correct is most useful in analysis tasks where accounting for batch effects is otherwise not straightforward, such as for visualizing the data or during unsupervised population structure discovery.”

My second major comment is that parts of the paper, the Results section in particular, are very hard to follow and would benefit greatly from editing for flow. If the reader works hard enough, he can figure out what the authors were thinking, but this kind of writing is not likely to make the reader positively disposed to the work. The authors should consider taking some of the standard advice about how to structure papers, paragraphs, and sentences. A good reference is “Ten simple rules for structuring papers,” by Mensh and Kording in PLoS CB (<https://doi.org/10.1371/journal.pcbi.1005619>).

We thank the reviewer for their input. We have followed the reviewer's suggestions to improve the flow of our Results section. These changes are detailed in individual responses below.

For example the first paragraph in "A statistical framework for meta-analysis of microbial community profiles" reads...

In this paragraph, I want to know what the method does and why (the "context" and "content" from the PLoS CB paper referred to above). How it is done is of secondary importance, as it will be described in detail in the "Methods" section. The first sentence is good. In the second sentence, I am expecting the context and content for the first component of MMUPHin. Instead, the sentence starts off with the implementation details ("based on linear modeling"). I might care about that information later, but in this context it's something that I have to carry around in my head while I wait for the context and content. Same goes for sentences three and four ("we combined well-validated data transformation and linear modeling combinations..." and "we generalized and formalized approaches from cancer transcriptional subtyping"), which is again telling us about implementation details when what we want are the context and the content. A version of this paragraph that would be less frustrating to the reader would be something like:

> "MMUPHin consists of three main components: batch and study effect correction, covariate modeling, and population structure discovery. To correct for batch and study effects, the first component of MMUPHin extends methods from the gene expression literature to give the user batch-corrected microbial abundance profiles. Second, to test for differential abundance while taking into account study and batch effects that remain after batch adjustment, we apply fixed- and random-effects models developed for meta analysis. Finally, to permit unsupervised discovery of discrete and continuous population structures, we generalize approaches from cancer transcriptional subtyping."

This has all the same information, but within each sentence the context comes first. This makes it easier for the reader to follow the logic: each sentence starts out with a problem and then describes the solution. Contrast with the initial version, in which the implementation details are described first. In that case, the reader is left with a lot of unresolved questions (why did you extend methods from the gene expression literature? Would something else be better?) that are not answered until much later.

A large number of paragraphs in the Results section suffer from similar issues.

We agree with the reviewer's input. The highlighted paragraph, along with other text discussing our methods, have been updated to describe context first before discussing the implementation details. These changes are too extensive to quote here, but please see the revised manuscript.

A related point that makes some of the paper difficult to follow is the fact that MMUPHin has many components, but the authors seldom specify which part of

the “collection of tools” they are referring to when they refer to MMUPHin. Again, this is something that the reader can figure out if he works hard enough, but the lack of precision is not very considerate. In particular, I often found myself spending time trying to figure out whether the authors were referring to results based solely on batch-corrected profiles or to results based on meta-analysis of batch corrected profiles (e.g. first paragraph of the section “Meta-analysis of the IBD microbiome”, the paragraph starting on line 115, the paragraph starting on line 126). As with many of my other comments, this is something the reader can figure out if he works hard enough, but it would be better for the authors to specify.

We thank the reviewer for this suggestion. To help the reader differentiate components of MMUPHin, we have systematically updated the manuscript text and display items, such that the four components are explicitly named as “MMUPHin_Correct” (batch correction), “MMUPHin_MetaDA” (meta-analyzed differential abundance testing), “Meta_Discrete” (discrete population structure discovery), and “Meta_Continuous” (continuous population structure discovery).

Some other notes:

- Line 91: This section is called “A statistical framework...” but it combines discussion of the statistical framework with discussion of the simulation studies. Discussion of the simulation should be in a separate section.

We thank the reviewer for their suggestion. We have added a separate section header to the simulation section: “Comprehensive validation of MMUPHin via realistic synthetic data”

- Line 95: “Covariate modeling” is very jargony.

We have changed this to “meta-analyzed differential abundance testing”.

- Line 126: This paragraph discusses both differential abundance testing and structure discovery. They should be split up unless you want the reader to spend time trying to figure out what the relationship between differential abundance testing and population structure discovery is.

We agree and have changed the paragraph break location so that differential abundance testing and population structure discovery methods are discussed entirely in two separate paragraphs.

- Line 144: My understanding is that this paragraph discusses what the batch-adjusted profiles look like and that the results are not based on the meta-analysis part of the pipeline. If this is not true the authors should clarify. If this is true, I am a little bit confused about why these profiles are being discussed in such detail. There are still substantial batch effects at this point that have not been dealt with, and so while there is some interest in the effect of the batch adjustment, we can't draw biological conclusions based on any of the phenomena described here.

The results discussed for the PERMANOVA analysis are indeed based on batch-corrected profile using MMUPHin_Correct, but not through the meta-analysis per-feature testing module (MMUPHin_MetaDA). However, they do not contain substantial batch effects that have not been dealt with per se. Instead, part of the goal of this discussion is to demonstrate the limitations of batch effect removal in real microbiome data, since it is incompletely eliminated by essentially any reasonable method. This is also true in our simulation data, and in other molecular data approaches that must deal with similar distributional effects (e.g. scRNA-seq).

In more detail, the purpose of this section is to report overall trends in the collection of microbial profiles that are affected either by batch differences or meaningful biological variables. While there are arguments for more microbiome-targeted approaches (e.g. MiRKAT [3]), the most ubiquitous one used for this is PERMANOVA. However, the orders by which variables enter the model affects PERMANOVA R2 results. As an example, in a dataset where batch has no effect on the microbiome, but is correlated with the biological exposure variable, “adjusting” for batch by including it in the PERMANOVA model before exposure will wrongfully diminish the effect of exposure, whereas entering it after the exposure will yield the same R2 for exposure as if batch was not adjusted for. This is a known property of PERMANOVA R2 [4], and the behavior would manifest in the same way in any model that makes similar assumptions about the priority of confounders. In light of this, to avoid unduly biasing our results in either direction, we elected to consistently report the “marginal” R2 for all tested technical and biological variables (i.e., each was tested univariably with PERMANOVA). While we agree that the effect sizes cannot exactly quantify batch-corrected biological signals, the difference in their orders of magnitude still provides an overall characterization of sources of variability in the data. Again, this practice is consistent with other previously published work in the field (e.g. [5]).

- Line 360: This paragraph is confusing because it says the same thing twice. The “In practice” is supposed to signal that the authors are starting the explanation over again from the beginning, but on the first read I thought that the sentence was expanding on just the metaforR part of the pipeline, not the whole thing. I’m sure the authors can come up with something better, but I would suggest instead of “In practice”, something like “Overall, for the meta-analysis part of the pipeline, the user provides...”

We thank the reviewer’s suggestion. To improve clarity, we decided to break up the paragraph into two parts, starting at “in practice”, so that the second paragraph is more clearly discussing MMUPHin_MetaDA as a whole. The new paragraph now reads:

“Overall, for running MMUPHin_MetaDA, the user provides...”

- Line 525: Is this paragraph complete? Did the authors want to describe the results of the computational cost study?

We apologize for omitting the discussion of computational cost here. The following has been added to the paragraph: “The computational cost of BDMMA is prohibitive when

compared to MMUPHin and quantile normalization, requiring ~5 total CPU hours to finish on the very moderately sized data (200 total samples by 200 features).”

References

1. Johnson WE, Li C, Rabinovic A: **Adjusting batch effects in microarray expression data using empirical Bayes methods.** *Biostatistics* 2007, **8**:118-127.
2. Ritchie ME, Phipson B, Wu D, Hu Y, Law CW, Shi W, Smyth GK: **limma powers differential expression analyses for RNA-sequencing and microarray studies.** *Nucleic Acids Res* 2015, **43**:e47.
3. Zhao N, Chen J, Carroll IM, Ringel-Kulka T, Epstein MP, Zhou H, Zhou JJ, Ringel Y, Li H, Wu MC: **Testing in Microbiome-Profiling Studies with MiRKAT, the Microbiome Regression-Based Kernel Association Test.** *Am J Hum Genet* 2015, **96**:797-807.
4. Tang ZZ, Chen G, Alekseyenko AV: **PERMANOVA-S: association test for microbial community composition that accommodates confounders and multiple distances.** *Bioinformatics* 2016, **32**:2618-2625.
5. Lloyd-Price J, Arze C, Ananthakrishnan AN, Schirmer M, Avila-Pacheco J, Poon TW, Andrews E, Ajami NJ, Bonham KS, Brislawn CJ, et al: **Multi-omics of the gut microbial ecosystem in inflammatory bowel diseases.** *Nature* 2019, **569**:655-662.